Corrected: Author Correction

# A standard for near-scarless plasmid construction using reusable DNA parts

Xiaoqiang Ma[1,2], Hong Liang[1,2], Xiaoyi Cui[1,2], Yurou Liu[1,2], Hongyuan Lu[1], Wenbo Ning[1], Nga Yu Poon[2], Benjamin Ho[1] & Kang Zhou[1,2]

Here we report GT (Guanin/Thymine) standard (GTS) for plasmid construction under which DNA sequences are defined as two types of standard, reusable parts (fragment and barcode). We develop a technology that can efficiently add any two barcodes to two ends of any fragment without leaving scars in most cases. We can assemble up to seven such barcoded fragments into one plasmid by using one of the existing DNA assembly methods, including CLIVA, Gibson assembly, In-fusion cloning, and restriction enzyme-based methods. Plasmids constructed under GTS can be easily edited, and/or be further assembled into more complex plasmids by using standard DNA oligonucleotides (oligos). Based on 436 plasmids we constructed under GTS, the averaged accuracy of the workflow was 85.9%. GTS can also construct a library of plasmids from a set of fragments and barcodes combinatorially, which has been demonstrated to be useful for optimizing metabolic pathways.

[1] Department of Chemical and Biomolecular Engineering, National University of Singapore, Singapore 119077, Singapore. [2] Disruptive & Sustainable Technologies for Agricultural Precision, Singapore-MIT Alliance for Research and Technology, Singapore 138602, Singapore. Correspondence and requests for materials should be addressed to K.Z. (email: kang.zhou@nus.edu.sg)

Biotechnology is transforming how humans generate fuels, produce chemicals, and treat diseases[1–3]. Developing the needed biotechnologies often requires construction of plasmid, a vector for carrying genetic information[4]. Currently, most researchers construct plasmids in a highly inefficient way— they customize genetic materials, pay commercial companies to synthesize the materials, wait for many days, and often only use them once. One way to improve the efficiency is to use standard DNA parts to construct plasmids, which has been explored but has not been adopted widely by the whole biotechnology community, possibly due to limitations in the previous designs. The most well-known standard of biological parts is BioBricks[5], which has been used since 2003 mainly by international Genetically Engineered Machine (iGEM)[6], a student competition in synthetic biology. Through the competition, BioBricks Foundation collects and distributes plasmids that carry standard parts, which can be combined to produce new standard parts with the help of restriction enzymes (REs). This system is less efficient in combining multiple DNA parts simultaneously—every round only two parts can be combined. In addition, scars will be generated when two DNA parts are connected, which could affect the biological function of DNA parts[7].

After the technologies that can assemble multiple DNA parts seamlessly were developed around 2009 (Gibson[8] and Golden Gate[9] methods), research labs swiftly adopted them to speed up projects. They have, however, mostly used customized parts till to date. In 2015, BASIC standard was developed[10], which allowed the use of standard DNA parts in multi-pieces DNA assembly, but it has so far not been used globally possibly due to some of its limitations, including low accuracy, difficulty in reuse of constructed plasmids, and leaving large scars. As a result, there was a need to develop a DNA assembly standard that can overcome all the aforementioned limitations and can also be compatible with most of popular DNA assembly methods.

Here, we report GT (Guanin/Thymine) DNA assembly standard (GTS) which has met the need, and has the potential to reduce the cost and time of plasmid construction in biotechnological applications if a considerable fraction of research community adopts GTS. GTS enables construction of plasmids in a near-scarless manner by using standard parts and is compatible with CLIVA, Gibson assembly, In-fusion cloning, and RE-based DNA assembly methods. GTS has an averaged accuracy of 85.9% based on 436 plasmids we constructed, and allows unprecedently flexible arrangements of reusable DNA parts. GTS is also suitable for constructing plasmid libraries that are very useful in many biotechnological applications, such as metabolic engineering.

## Results

**Principles and workflow of GTS**. GTS has three rules: (1) any DNA sequence longer than 35 nucleotides (nt), starting with G and ending with T can be defined as a fragment; (2) any DNA sequence between 20–80 nt can be defined as a barcode; (3) in plasmid construction any fragment must be placed after a barcode, and any barcode must be placed after a fragment. Since the first two rules can be easily satisfied, most functional DNA sequences can be defined as fragment and/or barcode (examples are provided in Fig. 1a). For example, promoter can be either fragment or barcode depending on its length (T7 promoter with *LacI* repressor expression cassette [~1.7 kbp] can be a fragment, while pJ23119 promoter with a length of merely 35 bp can be a barcode). To construct a plasmid, each end of a fragment is first connected to half of a barcode, which has a complementing half that is connected to another fragment. The barcoded fragments are then assembled into a plasmid in a specific order based on the pairing of barcode halves (Fig. 1a). This workflow always satisfies the third rule.

A basic requirement of any standard system is compatibility. In this context, it means any barcode can be placed after any fragment, which enables arrangement of fragments and barcodes in any order as long as the third rule is met. To have such compatibility, one has to conserve nucleotides at fragment ends to ensure their standard connections to any barcodes. A key innovation of this work is that we managed to minimize the length of conserved sequence at each end to be 1-nt, which is the minimal length of any conserved sequence.

Having longer conserved sequence constrains flexible use of fragments. For example, if a fragment is a protein coding sequence and the conserved sequence at its 3′ end is TAG (a stop codon), it is impossible to fuse a fluorescence protein to its C-terminus to study its cellular localization, because translation terminates at the stop codon. In GTS, the conserved sequence at 3′ end is T, which can be used to create 16 codons that start with T (including stop and nonstop codons) by connecting different barcodes to this end (Fig. 1b). This example also explains why we selected T as conserved sequence at 3′ end of fragment. Similarly, we chose G as conserved sequence at 5′ end of fragment, because it can be used to encode ATG (a start codon) or another 15 codons that end with G (Fig. 1b). An experimental validation of this flexibility is provided in Supplementary Fig. 1. Because of this flexibility, GTS almost eliminates scars, which often interfere with function of DNA parts by adding extra, undesired codons to coding sequence and/or incurring undesired DNA-protein interaction[7].

To connect halves of two barcodes to designated ends of a fragment (this process is defined as barcoding, the oligos used to create barcode halves are termed as Barcoding oligos [Boligos]), one needs to use DNA ligation techniques based on DNA sticky ends (SEs), which on fragment are derived from the conserved DNA sequences, and which on barcodes are added as standard connectors (Fig. 1c). Since the conserved sequence length is 1-nt in GTS, the corresponding SE length is 1-nt (Fig. 2a, b). A major challenge we faced was that DNA ligation based on such short SEs was inefficient, if we used the conventional method, in which two oligos were annealed to form one barcode half with a 1-nt overhang (Fig. 2a and Supplementary Fig. 2). We assessed ligation efficiency by using PCR: we attempted to amplify the barcoded fragments from the ligation product by using two Aoligos (Assembling oligos) that only bind the barcode halves—PCR would only be successful if enough SEs are ligated. In a test run with barcoding five fragments (Fig. 2a, c), although correct PCR products were observed on agarose gel after electrophoresis, undesired PCR products (in the form of smear) were also observed (Fig. 2a). When we further attempted to assemble the five barcoded fragments into a plasmid, we only obtained a few colonies and sequencing the plasmids they contained also revealed that a large fraction of the plasmids had duplication of some barcode halves (Fig. 2d, e). We tried the same procedure with another two plasmids, each of which was also assembled from five fragments, and we obtained similar, negative results (Fig. 2e and Supplementary Fig. 3a).

The duplication of barcode halves suggested that two barcode halves were ligated to one side of some fragments in a tandem manner. We hypothesized that if we sealed the blunt end of each barcode halve by connecting the two strands with a few extra nucleotides, this problem could be solved, because the barcoded fragments would be circular DNA molecule, leaving no end for additional barcode halve to attach (Fig. 2b). Essentially, each barcode halve would be created by using one oligo that has a stem-loop secondary structure (Figs. 1c and 2b). With this new design, we repeated construction of the same plasmids. We found that most undesired PCR products were eliminated (Fig. 2b). More importantly, the number of colonies we obtained from the

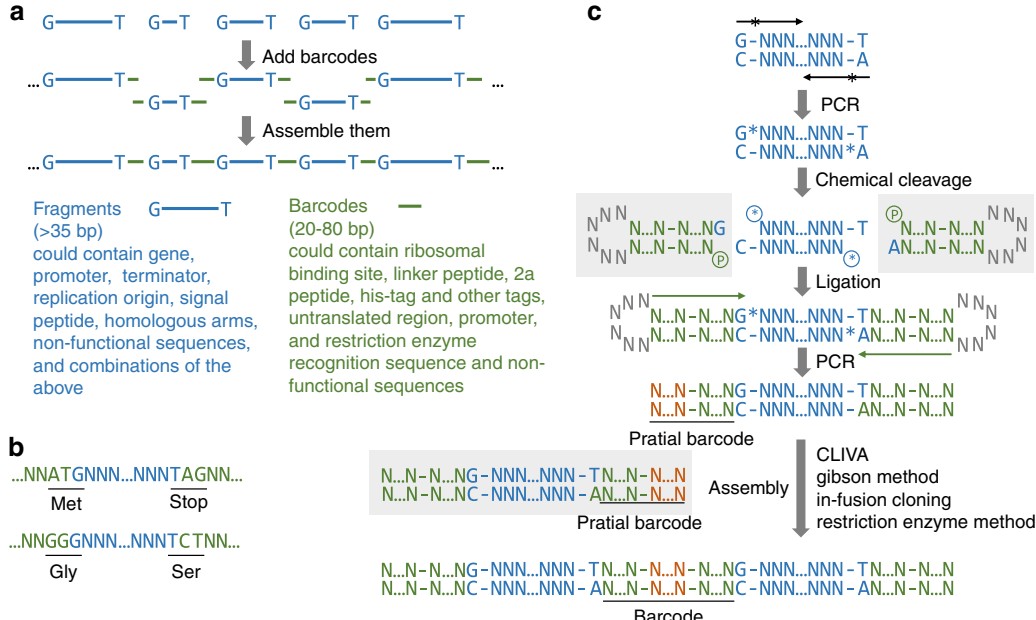

**Fig. 1** GT assembly standard (GTS) for constructing plasmid from standard parts (fragments and barcodes). **a** The plasmid construction workflow and examples of fragments and barcodes; **b** Using minimal conserved sequence (one nt long, G and T) in fragment eliminated scar in most DNA assembly practices. Two scarless connections are shown here. **c** Mechanism of adding two partial barcodes to one fragment, and mechanism of using two partial barcodes containing long overlapping sequence or restriction enzyme cutting sites (highlighted in orange N...N) to assemble two fragments by using various DNA assembly methods. * Indicates phosphorothioate (PS) bond; * in a circle indicates PS group; — indicates phosphodiester bond; P in a circle: phosphate group; ... indicates omitted, unspecified nucleotides. Items in gray shade are new items introduced in the step. Long black arrows are the oligos (termed as Fragment oligos [Foligos]) used to amplify fragment, and long green arrows are the oligos (termed as Assembling oligos [Aoligos]) used to amplify barcoded fragment. The oligos used to prepare barcode halve with stem-loop structure are termed as Barcoding oligos (Boligos)

subsequent plasmid construction was ~100 fold higher than that from the conventional method (Fig. 2d), and the plasmids these colonies contained were confirmed to be correct by sequencing in all our tests (Fig. 2e and Supplementary Fig. 3b).

In this workflow, fragments with a length over 59 nt were usually amplified from template DNA by using Fragment-creating oligos (Foligos, Fig. 1c). Posphorothioate (PS) bonds were introduced into proper locations of fragment by using PS-modified Foligos during PCR, and can be cleaved by using an Iodine-based chemical reaction[11] to generate the 1-nt SEs (Fig. 1c). In case PS-modified oligos are not preferred, one can use type IIS RE and non-modified oligos to create the fragment with the 1-nt SEs (elaborated in the next section of "Results"). Fragments with a length between 36 and 59 nt can be generated by annealing two single strand oligos directly, and this process would add the 1-nt SEs to fragments because of the oligo design (Supplementary Fig. 4). These oligos are termed as Noligos (Non-modified oligos for creating fragments). Barcoded fragments are amplified by a set of Aoligos that bind the barcodes and the PCR products can be assembled into a plasmid. To highlight that we developed a standard, NOT a DNA assembly method, we have demonstrated that we can use many popular DNA assembly methods in this step (Fig. 1c). When we used CLIVA, Gibson and In-fusion method (the results of testing these three methods under GTS can be found in Supplementary Fig. 5), the barcode regions of barcoded fragments were used to generate long SEs (CLIVA and Gibson), or were utilized as homologous regions (In-fusion). When we used conventional RE method, we included their recognition sequences in the barcode regions (Fig. 1c, elaborated in the last section of RESULTS).

We have successfully constructed 436 plasmids by using this workflow and CLIVA as assembly method (Supplementary Fig. 6a) for various projects from an expanding library of fragments and barcodes (Supplementary Data 1 and 2). The constructed plasmids with versatile functions can be used for genetic engineering of multiple microbial hosts (Supplementary Fig. 6b). The mean of assembly accuracy was 85.9% based on a two-step validation process (colony PCR followed by Sanger sequencing), and the assembly accuracy did not substantially change when plasmid length (2.43–13.04 kb) and the number of fragments used in plasmid construction (2–7) varied (Fig. 3). The median of the assembly accuracy was 100% (Supplementary Fig. 6c). Seventy-two percent of the identified errors were mutation/deletion/insertion within fragments, which were caused by PCR (Supplementary Fig. 6d). This large-scale experimental validation proved that the workflow of GTS is reliable and robust. In comparison, the averaged accuracy of BASIC method was only 50% when six fragments were assembled and one antibiotic was used[10].

**Using Type IIS REs to generate fragment with 1-nt SEs**. One limitation of GTS was the use of PS-modified oligos, which are more expensive and have longer synthesis time than non-modified oligos. We thus developed a method to generate the fragment with the 1-nt SEs by using non-modified oligos. We experimentally verified that a panel of type IIS REs could be used to replace the chemical cleavage after their recognition sequences are included in non-modified Foligos (Fig. 4a, b). We tested a one-pot reaction (RE-digestion and ligation) to barcode seven fragments, and barcoded fragments could be amplified by using PCR (Fig. 4c). The obtained PCR products were successfully assembled into three plasmids (Supplementary Fig. 7), suggesting that the 1-nt SEs on both ends of a fragment were produced efficiently in the one-pot reaction. Five REs can be used to generate such 1-nt SEs, so any of them can be used with a fragment if

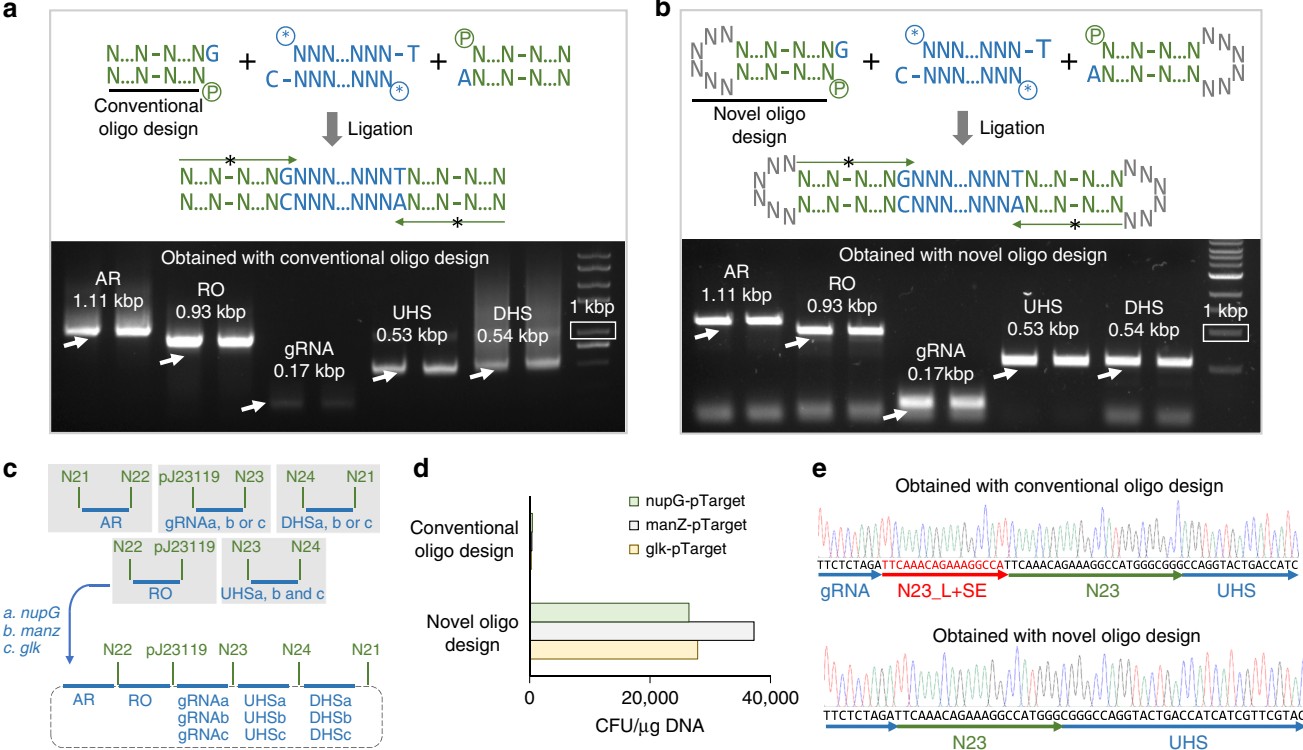

**Fig. 2** Development of a near-scarless barcoding method. Workflow of barcoding a fragment by using conventional **a** and novel **b** oligos designs. After the ligation, barcoded fragments were amplified by using two Aoligos (shown as long black arrow with *) that bind barcodes. Analysis of PCR products from **a** and **b** using gel electrophoresis are shown. White arrows indicate the desired bands, and one sample was loaded into two lanes. DNA marker with 1 kbp size is highlighted in white rectangular box. The lane symbols are explained below. **c** The three plasmids used to compare the two barcoding methods. Each of the three plasmids can be used to delete a gene in *E. coli* (*nupG*, *manZ* or *glk*) by using CRISPR/Cas9 technology, and was constructed from five fragments (blue thick horizontal bars represent fragments): antibiotic resistance marker (AR: Spec[R]), replication origin (RO: pMB1), guide RNA (gRNA), upstream homologous sequence (UHS), and downstream homologous sequence (DHS). Green texts indicate barcode; N21, N22, N23, and N24: non-functional connectors; pJ23119: a constitutive promoter. Five barcodes were used in each plasmid construction. PCR test results of the other two plasmids are shown in Supplementary Fig. 3a. **d** Colony forming unit (CFU/μg DNA) of three plasmids constructed by using fragments that were barcoded by conventional and novel oligo design. **e** Representative sequencing results. Two plasmids in each plasmid construction were sequenced. All the plasmids constructed by using novel oligo design were sequenced to be correct (6/6). Half of the plasmids constructed by using conventional oligo design were found to have assembly errors (3/6). A pair of sequencing results from the *manZ* plasmid construction are shown here. The other sequencing data can be found in Supplementary Fig. 3b. The red arrow and text indicate the insertion to the sequence. The functionality test of the constructed plasmid of *nupG*-pTarget was presented in Supplementary Fig. 18

the fragment does not have an internal recognition sequence of the RE. We analyzed the existing 332 fragments in our lab, and found that 297 of them (89.5%) can be barcoded by at least one of the five REs. The internal cutting sites in those fragments that have five REs' site can be removed through silent mutagenesis, just as what is being done with Golden Gate assembly.

**Flipping standard parts in plasmid construction.** A unique feature of GTS is that any fragment and barcode can be flipped in plasmid by using standard parts (Boligos and Aoligos). Each fragment can be prepared by using two Foligos that had PS bond after their first nucleotide at 5′ end (Figs. 1c, 2b and 5a). Each barcode in GTS has the following structure: L-SE-R (Fig. 5b). SE is the region that encodes the long (15–20 nt) Sticky Ends, or homologous regions (they can also contain RE sites); L and R are the regions flanking Left and Right side of SE, and both are allowed to be empty. The two halves of each barcode have the following structures: L-SE and SE-R. Each Boligo encodes one barcode half. Because each barcode half may be connected to either end of a fragment, there are two Boligos for each barcode half, carrying G or A overhang (Fig. 5c). Each barcode thus has four Boligos and four Aoligos (Fig. 5c, d). There are eight possible

ways to link two fragments through one barcode, as each fragment and the barcode can be flipped independently (Fig. 5d). The combinations of Boligos to implement the eight connection types are listed in Fig. 5e.

We experimentally demonstrated the eight connection types by flipping replication origin (RO: pMB1), antibiotic resistance marker (AR: SpecR) and the barcode connecting them in a plasmid that can express green fluorescence protein (GFP, Fig. 5e) in *Escherichia coli*. The plasmids were sequenced to ensure that the desired arrangements were achieved. Interestingly, the expression level of GFP varied substantially (up to close to four-fold change) among the eight constructs (Fig. 5f), proving the importance of being able to flip elements of plasmid. The difference could be caused by the presence of known and hidden promoters in AR and RO or collision of DNA polymerase (used for plasmid replication) and RNA polymerase (used for transcription) on plasmid[12]. These hypotheses are good topics for future studies for better controlling protein expression. Flipping parts also allows arranging standard parts in ways that are more complex and are more similar to how DNA parts are naturally arranged in genome[13], e.g., arranging two operons on sense and antisense strands to avoid transcriptional crosstalk, and

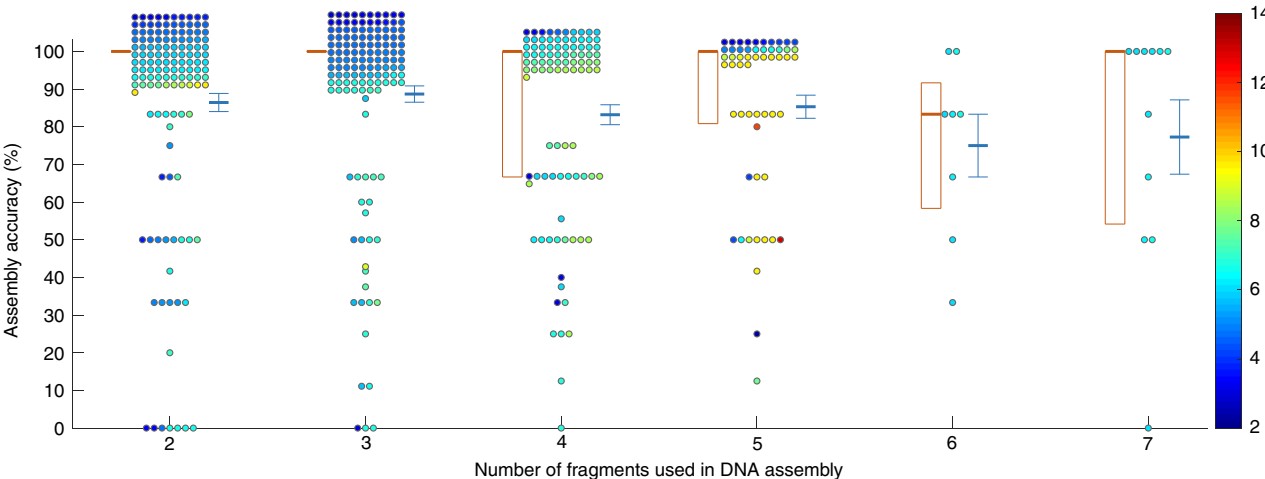

**Fig. 3** Statistics of 436 plasmids constructed under GTS. Assembly accuracy was percentage of colonies that have the correct plasmid. Each circle presents one plasmid construction. Face color of each circle indicates length of the plasmid (the color bar is provided, unit: kb). Each thick orange horizontal bar presents median of accuracies of a group (plasmids constructed from the same number of fragments were included in one group). Each orange box indicates 1st and 3rd quantiles of accuracies of a group. Each thick blue horizontal bar and the related error bar indicate mean and standard error of accuracies of a group respectively. For illustration purpose, circles with the same accuracy and fragment number are distributed around accurate values of accuracy and fragment number on the plot

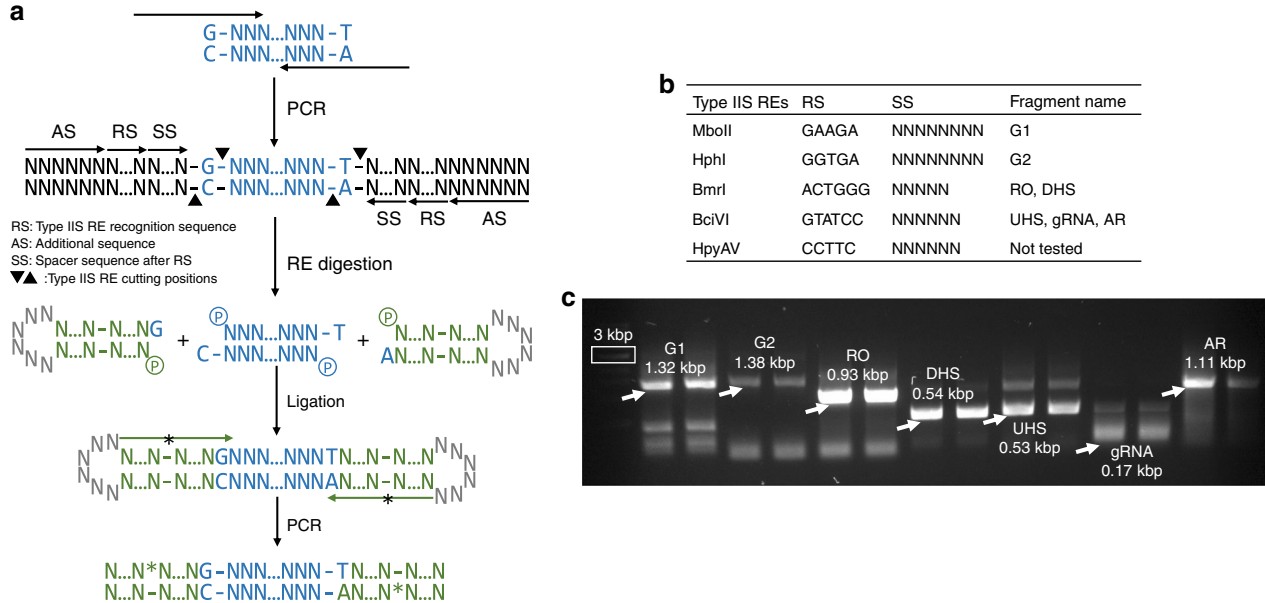

**Fig. 4** Development of Type IIS RE-based barcoding method. **a** Principal of using Type IIS RE to generate fragment with 1-nt SEs. The non-modified Foligos (shown as long black arrow) include additional sequence (AS, six random nucleotides), recognition sequence (RS), spacer sequence (SS, the length of SS depends on Type IIS RE used) and primer binding site. The amplified fragments were enzymatically digested by using the corresponding REs, and ligated with two Boligos. The ligation mixtures were used as templates in ligation PCR to amplify barcoded fragments by using corresponding PS-modified Aoligos (shown as long green arrow with *). **b** Five Type IIS REs can be used to generated 1-nt SEs. We have tested four of them to barcode seven fragments. The RS and SS of the five Type IIS REs are provided. G1: gene 1; G2: gene 2; RO: replication origin (pMB1); DHS: downstream homologous sequence; UHS: upstream homologous sequence; gRNA: guide RNA; AR: antibiotic resistance marker (Spec[R]). **c** Analysis of the products of ligation PCR. The desired bands are indicated by white arrows. Five barcode fragments (RO, DHS, UHS, gRNA, and AR) could be assembled into plasmid *nupG*-pTarget (Fig. 2c). DNA marker with 3 kbp size is highlighted in white rectangular box. The other two barcoded fragments (G1 and G2) were cloned into a pre-prepared plasmid backbone to construct two plasmids (Supplementary Fig. 7a). The transformation efficiencies and assembly accuracies of these three plasmids can be found in Supplementary Figs. 7b and c. All the three plasmids were constructed accurately based on sequencing results

placing two repeated sequences on the two strands to avoid undesired homologous recombination.

**Modularly constructing and modifying plasmids under GTS.** A few examples are provided to illustrate the modularity of GTS for constructing and editing plasmid. The first demonstration involved the construction of 16 plasmids for engineering *E. coli* to overproduce tyrosine (Fig. 6a–c), a valuable aromatic amino acid[14]. Each plasmid was assembled from six or seven fragments, which included one promoter, two genes, one terminator, one RO

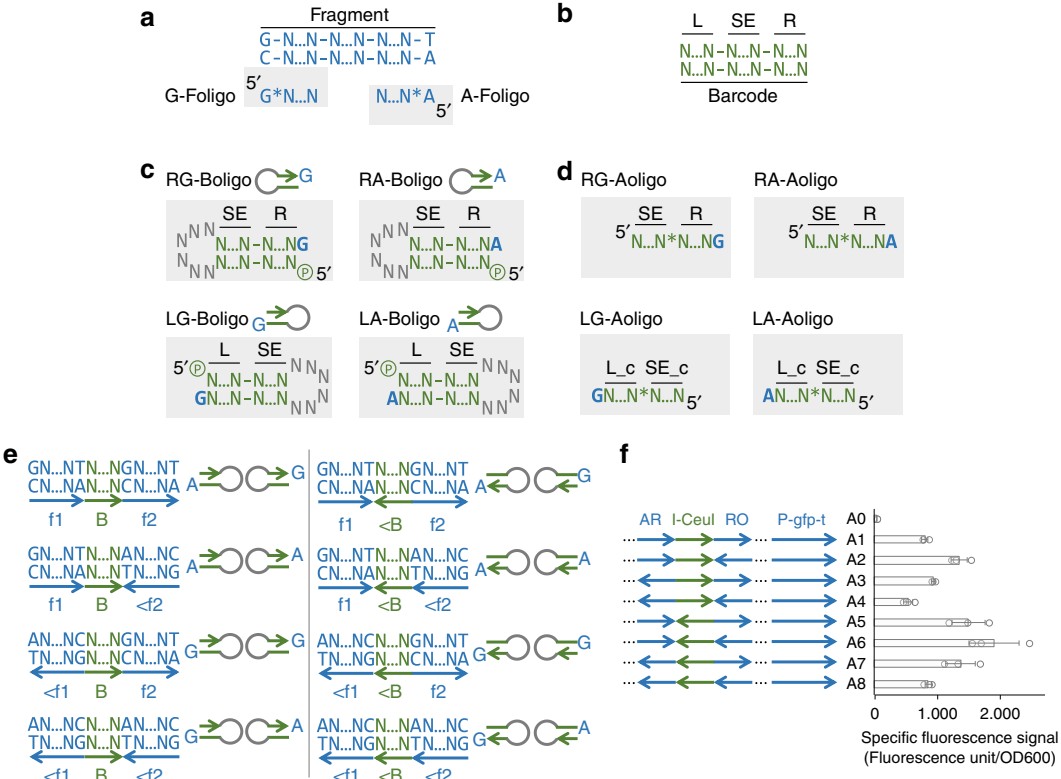

**Fig. 5** Principles of designing and using the oligos under GTS. **a** PS-modified Foligos are used to amplify fragment by using PCR. Position of each PS bond is shown. **b** Structure of barcodes. Any barcode is composed of L, SE, and R. L and R can be empty. SE should be 15–20 nt depending on its melting temperature and should not contain any sequence that may result in self-ligation. **c** There are four Boligos to encode all the variants of a barcode. The six nucleotides (NNNNNN in gray) can be any sequence as long as the illustrated stem-loop structure can be formed. The arrow in Boligo indicates the strand on which the barcode half is encoded. The length of Boligo is no longer than 90 nt. Each Boligo has a corresponding Aoligo. * Indicates a PS bond; − indicates a phosphodiester bond. There is one more PS bond at the center of each SE (not shown for simplicity), which is used to improve DNA assembly efficiency. **d** Instructions for connecting two fragments (f1 and f2) through one barcode (B) in eight ways. Prefix "<" indicates that a part is flipped; The blue arrows below sequences indicate fragment's orientation, while green arrows indicate barcode's orientation. The pair of Boligos used in each way is shown at the right side of each sequence. The nucleotides (G and A) used in fragment-barcode ligation are shown in bold font. **e** Demonstration of the eight connection options in a simple example. AR: antibiotic resistance marker (SpecR); I-Ceul: a barcode containing a 19 bp RE recognition site; RO: replication origin (pMB1); P: promoter; gfp: a gene encoding green fluorescence protein; t: terminator. **f** Each row indicates a plasmid (A1–A8). Arrow indicates the orientation of fragment/barcode. Only relevant parts of each plasmid are illustrated. Each plasmid's performance (i.e., specific fluorescence signal) is shown in the chart. Empty circles indicate values of replicates. Each bar indicates the mean of triplicates at each condition. Empty plasmid (A0) without gfp was used as a negative control. Error bars indicate standard error ($n = 3$)

and one AR. There were four RO candidates, two ways of connecting the genes, and two promoter candidates, so there were 16 plasmids in total (4x2x2, Fig. 6a, b, d, Supplementary Fig. 8). Seven barcodes were used. Three of them were functional—two containing ribosomal binding sites (RBSs) and one providing stop codon (3UTR)—and the rest merely served as connectors. The E. coli strains harboring the 16 plasmids exhibited a wide range of abilities in producing tyrosine (Fig. 6d). These plasmids constructed modularly under GTS provide a basis for further demonstrations.

Using standard parts, we can replace fragments in, remove fragments from, and add fragments to plasmid constructed under GTS (Fig. 6e, f, g). Below are three demonstrations. We replaced the promoter (PT7) of the best-performing plasmid (among the 16 plasmids; plasmid name: TPP2) with PthrC3, a short auto-inducible promoter[15]. We amplified the whole plasmid except PT7 by using two Aoligos of the barcodes that flanked PT7, barcoded PthrC3 with the same barcodes, and assembled them into one plasmid (Fig. 6e). The new plasmid (TPP17) led to slightly higher tyrosine titer than the one with PT7 (Fig. 6d, h), and simplified the fermentation process by eliminating addition

of inducer (PT7 requires addition of Isopropyl β-D-1-thiogalactopyranoside as an inducer). To test if there is any hidden promoter upstream of PT7, we needed to remove PT7 from plasmid TPP2 (Fig. 6f). We amplified the plasmid except PT7 and an adjacent fragment (RO) by using two Aoligos of the barcodes that flanked them, barcoded the adjacent fragment (RO) with the same barcodes, and assembled them (Fig. 6f). Surprisingly, the constructed plasmid (TPP18) still led to production of $1 \, g \, L^{-1}$ of tyrosine (Fig. 6h), indicating presence of hidden promoter(s) in the upstream sequence. We had the parental strain with an empty plasmid as the negative control (TPP0, a plasmid without the coding sequences), which was confirmed not to produce tyrosine (Fig. 6h). To test the theory of hidden promoter, we replaced PthrC3 with a terminator. The titer of tyrosine was substantially reduced to $0.33 \, g \, L^{-1}$ with the new strain (data not shown), supporting the hidden promoter hypothesis. Tyrosine can be deaminated into p-coumaric acid, a precursor of many valuable flavonoids[16], when gene tal (encoding tyrosine ammonia-lyase) was expressed[17] (Fig. 6c). When we added tal into TPP17 by using standard oligos as described in Fig. 6g, E. coli strain with this new plasmid (PCAP1) readily produced

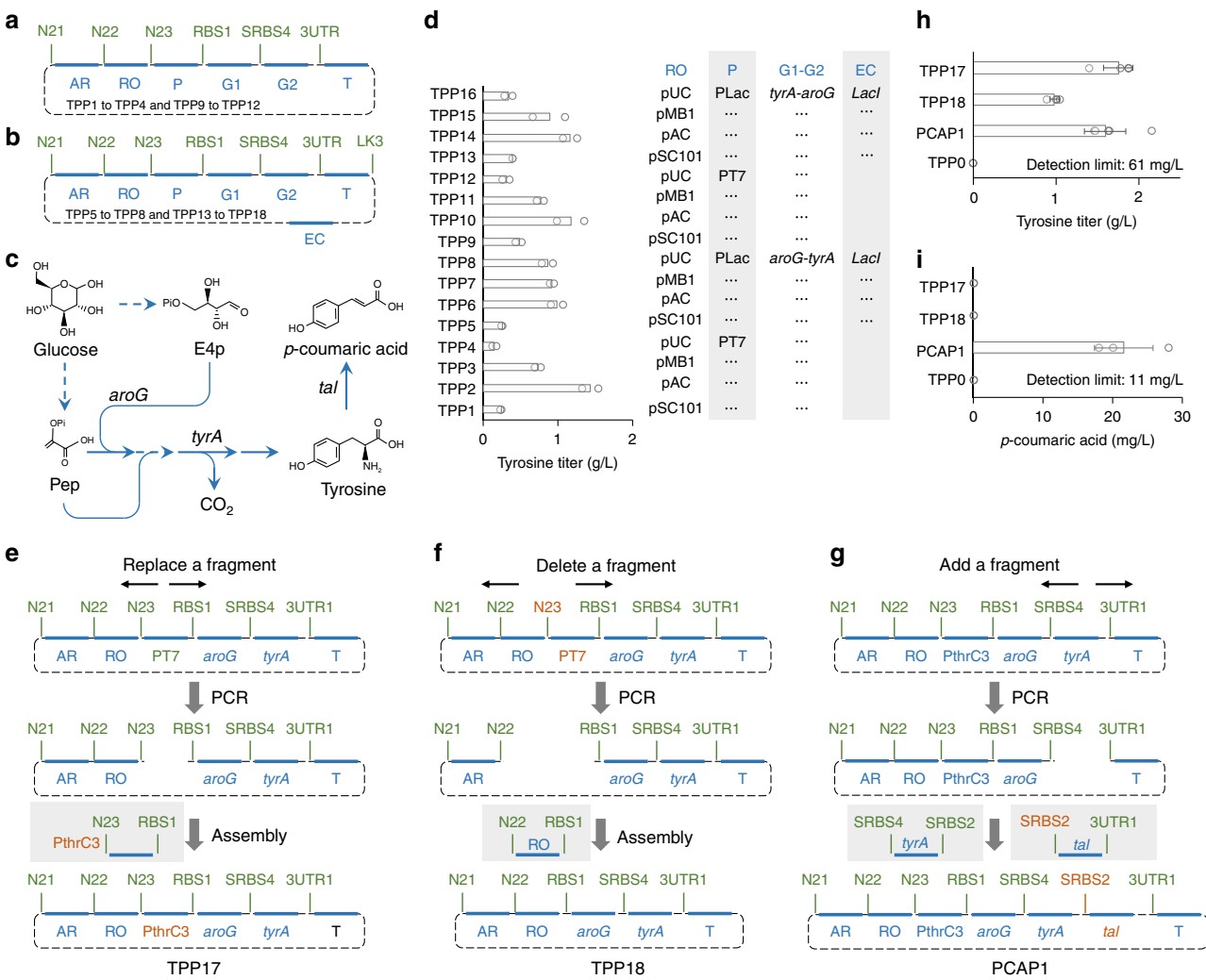

**Fig. 6** Constructing and editing plasmids using standard parts under GTS. Structure of the 16 plasmids for improving tyrosine production in *E. coli*, **a** Six- or **b** seven-fragment assembly were performed to construct these 16 plasmids (Supplementary Fig. 8). Blue thick horizontal bars represent fragments; RO: replication origin (pSC101, pAC, pMB1, or pUC); P: promoter (PT7 or PLac); G1, G2: two genes in an operon (*tyrA-aroG* or *aroG-tyrA*); T: terminator; AR: antibiotic resistance marker (Spec$^R$); EC: *LacI* repressor expression cassette. Green texts indicate barcodes; N21, N22, N23, and LK3: non-functional connectors; RBS1: ribosome binding site; SRBS4: stop codon and ribosome binding site; 3UTR: stop codon and untranslated region. **c** The biosynthetic pathway of tyrosine and *p*-coumaric acid from glucose. Pep: Phosphoenolpyruvate; E4p: Erythrose 4-phosphate; *aroG*: a gene encoding mutated *E. coli* 3-deoxy-7-phosphoheptulonate synthase; *tyrA*: a gene encoding mutated *E. coli* fused chorismate mutase/prephenate dehydrogenase; *tal*: a gene encoding tyrosine ammonia-lyase from *Saccharothrix espanaensis*. **d** Composition of the 16 plasmids and their corresponding tyrosine titer. TPP1-16: Tyrosine-Producing Plasmid 1–16, created by combination of RO, P, and G1–G2 as indicated. Empty circles indicate values of replicates. Each bar indicates the mean of duplicates at each condition. **e–g** Editing a plasmid by replacing, deleting, or adding a component by using standard oligos. Orange texts indicate the component changed and added. Thin black arrows indicate the PS-modified Aoligos used in PCR. **h, i** Characterization of strains carrying the three plasmids obtained from (**e–g**) in terms of tyrosine (**h**) and *p*-coumaric acid **i** production. Empty circles indicate values of replicates. Each bar indicates the mean of at least three replicates at each condition. Error bars indicate standard error ($n >= 3$)

22 mg L$^{-1}$ of *p*-coumaric acid (Fig. 6i), while the strains carrying the plasmids without *tal* (TPP18 and TPP0) did not produce a detectable titer of *p*-coumaric acid (Fig. 6i).

As shown in the above experiments, plasmids were often improved through multiple rounds of modifications, because plasmid performance needed to be assessed experimentally and used as feedback to direct the next round of plasmid construction. GTS is the first standard that allows editing plasmids by using standard parts.

**Constructing plasmid library to optimize metabolic pathways.** If a barcoded fragment in plasmid construction was replaced by a mixture of fragments that were barcoded in the same way, a mixture of plasmids would be obtained. If two barcoded fragments were

replaced in this way, a more diverse mixture of plasmids would be obtained. Such plasmid mixture is termed as plasmid library, and can be used for combinatorial optimization of metabolic pathway. To demonstrate this concept, a small plasmid library was built for improving *E. coli*'s ability of producing *p*-coumaric acid (Fig. 7a and Supplementary Fig. 9). We first constructed six plasmids that covered all possible ways of shuffling *aroG*, *tyrA*, and *tal* (the three genes we used for *p*-coumaric acid production) in an operon (Fig. 7a, b). The operon together with its promoter and terminator is Module 1 (M1). We further selected another three genes that were reported to improve production of aromatic compounds (*tktA*, *aroE* and *ppsA*, Fig. 7b), and constructed six plasmids to shuffle them in an operon, which together with its promoter and terminator is Module 2 (M2). Variants of M1 and M2 can be easily

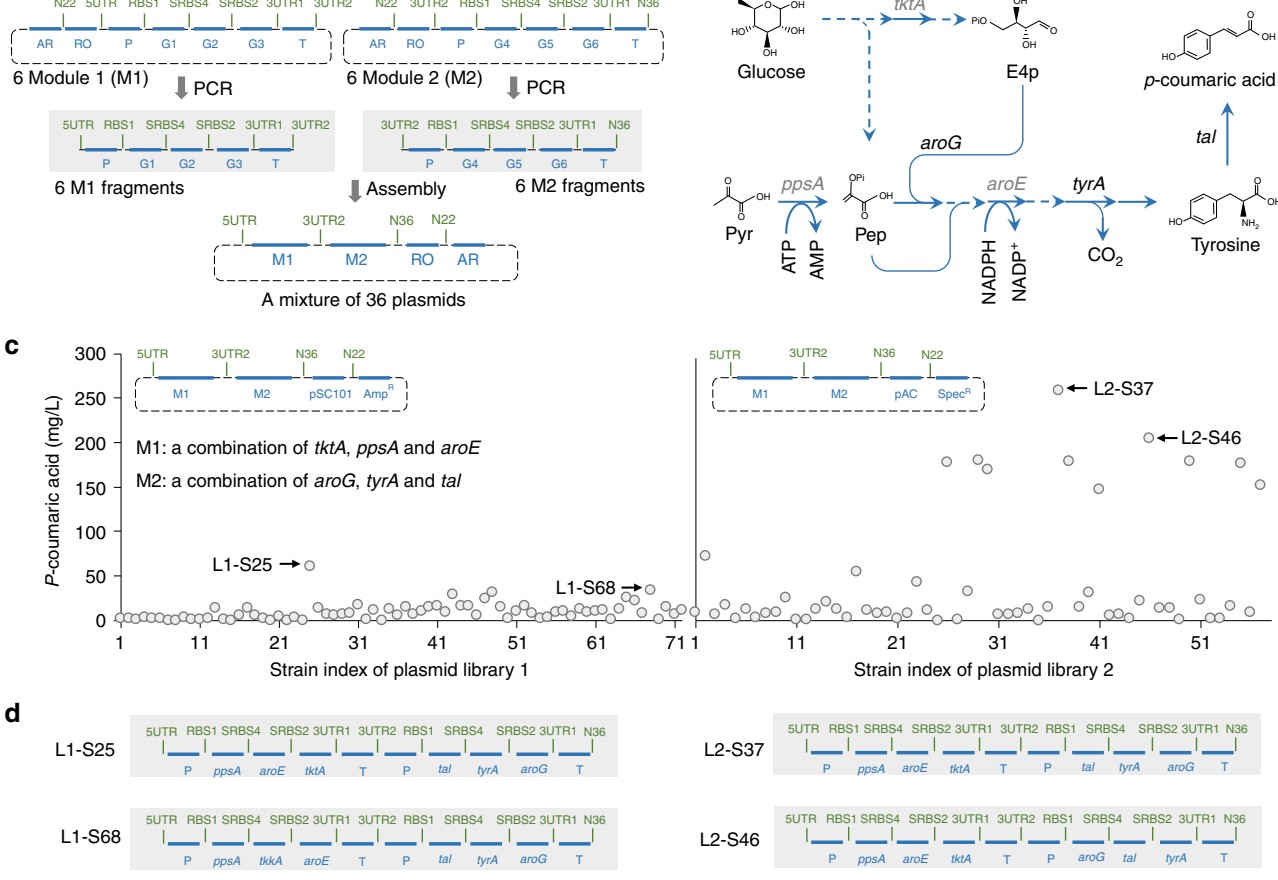

**Fig. 7** Constructing plasmid library using standard parts for combinatorial optimization of *p*-coumaric acid production in *E. coli*. **a** Construction of two combinatorial plasmid libraries to enhance *p*-coumaric acid production of *E. coli*. Six genes involved in shikimate production (*ppsA*, *tktA*, and *aroE*) and *p*-coumaric acid production (*aroG*, *tyrA*, and *tal*) were divided into two modules. Each module has six variants resulting from shuffling three genes. To prepare one library, six Module 1 (M1) plasmids and six Module 2 (M2) plasmids were made. Both M1 and M2 plasmids used the same promoter (pthrC3) and terminator (T7 terminator). Then, six M1 fragments and six M2 fragments were amplified from M1 and M2 plasmids by using two sets of Aoligos as indicated in black arrows (PCR results are presented in Supplementary Fig. 9a). The amplified M1 and M2 fragments were mixed equimolarly, and assembled with a plasmid backbone (pSC101 with Spec[R] or pAC with Spec[R]). In library 1, pSC101 was used. In library 2, pAC was used. More details of the library construction and the quality control data are described in Supplementary Fig. 9b. **b** Extended biosynthetic pathway of *p*-coumaric acid from glucose. Pyr: Pyruvate. *ppsA*: a gene encoding of *E. coli* phosphoenolpyruvate synthetase; *tktA*: a gene encoding of *E. coli* transketolase; *aroE*: a gene encoding of *E. coli* shikimate dehydrogenase. **c** Screening the two plasmid libraries. The mixture of plasmids was used to transform an *E. coli* strain (MG1655_ΔrecA_ΔendA_ΔpheA_ΔtyrR_DE3). For each library, seventy-two colonies (two times library size) were screened with *p*-coumaric acid titer as evaluation metric. Note that only a fraction of colonies from library 2 can be successfully cultured in liquid medium (57/72). Each circle indicates *p*-coumaric acid titer of a strain. **d** Top 2 strains from **c** were characterized by sequencing the corresponding plasmids. The arrangement of the genetic parts of the corresponding plasmids is presented

amplified from the 12 plasmids and combined into new plasmids (Fig. 7a). Two libraries were created. Each contained full combinations of six variants of M1 and M2 (expected library size: 6 × 6 = 36). The difference between the two libraries was the plasmid copy number (Fig. 7c). We picked 72 colonies (two times the size of the library) after transformation of *E. coli* with each plasmid library, and screened them based on the titer of *p*-coumaric acid. In general, the library with higher copy number replication origin (Library 2) produced more *p*-coumaric acid, and the top performer produced 263 mg L$^{-1}$ of *p*-coumaric acid (Fig. 7c), which was more than 10 times higher than that before this combinatorial optimization (Fig. 6i). We determined the identity of the plasmids responsible for the higher *p*-coumaric acid production by sequencing (Fig. 7d).

In many biotechnology applications, due to lack of accurate in silico models and/or in-depth mechanistic understanding of the biological systems, combinatorial optimizations have been widely used and proven to be effective[18, 19]. In this study, we did not intend to achieve highest tier of tyrosine and *p*-coumaric acid

(reports with higher values are available[20–22]), instead, we used these optimization examples to demonstrate the unique features of GTS in a relevant context.

**Constructing and editing plasmids using REs**. We used a two-tier assembly process when we constructed the plasmid library. In the second-tier assembly, we amplified the expression modules by using PCR, which could introduce undesired sequence changes to the modules. To overcome this drawback, we designed barcodes that contain RE sites, and have demonstrated that RE-based methods can be used in the second-tier tier assembly without using PCR.

The recognition sequence (RS) of various types of RE can be included in the barcodes so Golden Gate method[9] or conventional RE-based cloning could be employed in the second-tier tier assembly as long as RE sites are absent in the fragments (Supplementary Fig. 10a and 10b). For example, we incorporated

BsaI's RS and spacer into three barcodes we used to construct the plasmid libraries. With such design, we were able to assemble M1 and M2 into the destination plasmid efficiently and accurately without using PCR (Supplementary Fig. 10c to 10g).

By adopting barcodes that contain RE sites, we were also able to modify the constructed plasmid over 10 kb without amplifying the whole plasmid. As a demonstration, we constructed one plasmid carrying the two modules (M1 and M2) (Supplementary Fig. 11a and 11b) by using three barcodes (5UTR, REC1 [containing three 8-bp RE recognition sites] and I-CeuI [containing a long 19-bp RE recognition site]). The three barcodes allowed replacement of any module with a variant by using REs. For example, if the fragment of *tal* in M2 needs to be removed (Supplementary Fig. 11a), the M2 can be replaced with a variant (M3) that does not contain *tal* by using the conventional RE-based cloning as the barcodes of REC1 and I-CeuI on both ends of M2 and M3 contain the needed RE recognition sites (Supplementary Fig. 11c and 11d). We have experimentally validated this proposed workflow (Supplementary Fig. 11e to 11i).

With REs, it is possible to use standard parts to modify many (commercial) plasmids that have been constructed prior to GTS. For instance, a first-tier plasmid (RECP1) carrying a three-gene operon could be easily constructed by using standard parts and CLIVA (Supplementary Fig. 12). The plasmid had two barcodes flanking the three genes, and the two barcodes (REC3 and REC4) were derived from the multiple cloning sites of pHT01 (a protein expression plasmid used in *Bacillus subtilis*). The three-gene operon can be liberated by using two REs (BamHI/AatII), and cloned into similarly digested pHT01 (Supplementary Fig. 12). By incorporating specific RE sites into barcode, it is very flexible to select suitable RE for the insert (single or composite DNA fragments) from a large collection of RE-containing barcodes. Through a set of GTS barcodes that contain various REs and RBSs, any combination of a few genes can be overexpressed as an operon by using any conventional expression vectors that use RE-based cloning. RE has been widely used for constructing plasmids throughout the past decades, and thus, a large number of plasmids with versatile biotechnological applications have been constructed and can be utilized by this workflow. We hope to emphasize that if we need to clone the three-gene operon into another vector by using different REs, we would not need to order new oligos for the three genes. We only need to use another two barcodes that contain the needed REs. Because barcodes are also standard parts, we would not need to order any customized parts to implement the plasmid construction.

## Discussion

We did not deliberately select the combination of the barcodes when we constructed the 436 plasmids—we simply chose/designed those that were needed by the application. The fact that most of them worked suggested that most barcodes we have designed were compatible with each other. All these plasmids were constructed by using CLIVA, which assemble DNA fragments through long SEs (15–20 nt). Theoretically, as the number of fragments increases, the likelihood of undesired self- and cross-annealing of SEs of barcodes would be increased. To further test if most barcodes are compatible with each other, we used three sets of barcodes with the same 7-fragment assembly (Supplementary Fig. 13). Good assembly efficiency and assembly accuracy (over 50%) were achieved by using two sets of barcodes, further supporting good compatibility of GTS barcodes.

A barcode can have multiple functions. For instance, LK3, a barcode that was used as non-functional barcode for construction of plasmid TPP5 (Fig. 6b), can also be used as functional barcode to create fusion protein (Supplementary Fig. 4b); 5UTR (5′

untranslated region), a barcode that was used to construct a plasmid expressing protein in *Saccharomyces cerevisiae* (Supplementary Fig. 14), was used as a non-functional connector for constructing the plasmid libraries (Fig. 7a). These practices increased the reusability of barcode-associated oligos, whose cost would be reduced when they are used in more plasmid constructions. Based on the analysis of constructing the 436 plasmids, each barcode-associated oligo has been reused 14 times on average (Supplementary Fig. 15).

In many cases, plasmid construction often requires short genetic elements (e.g., guide RNA of CRISPR-Cas9 system, and intron of *S. cerevisiae*). Under GTS, the needed plasmids can be easily constructed by using CLIVA which can precisely control the length of long SEs for short fragments (Supplementary Fig. 4b), while it is difficult for the methods that rely on exonucleases (e.g., Gibson[8] and SLIC[23]).

With the provided demonstrations in this study, we hope to highlight again we have developed a new DNA assembly standard with an open and flexible architecture (Supplementary Fig. 16) instead of a DNA assembly method. The key advantage of GTS is being able to efficiently ligate two barcode halves with any fragment through 1-nt SEs, allowing users to flexibly define the barcode sequence with or without biological functions, and select the appropriate DNA assembly methods based on their preference and the requirements of plasmid construction.

Pathway engineering is an important strategy to improve yield and productivity of microbial fermentation processes that were designed to produce valuable metabolites[24]. GTS enables constructing operon-based plasmid library by shuffling all the related genes in an operon by using standard parts. GTS can construct plasmid for pathway engineering in various microbes. We have described the construction of plasmids for *E. coli* and *B. subtilis* above, and we believe that GTS also enables constructing plasmids for pathway engineering in *S. cerevisiae*. As a proof-of-concept, we constructed three plasmids in *E. coli* and further assembled them into a larger plasmid by using fragments as homologous arms in *S. cerevisiae* (Supplementary Fig. 14). *S. cerevisiae* is well known for its homologous recombination activities. It has been used extensively in construction of large plasmids[24] and has been engineered as cell factory to make many valuable chemicals[25]. The simple demonstration shows that GTS should be able to provide standard parts for these efforts with the yeast, which should reduce time and cost of the related projects. Collectively, the features of GTS that have experimentally verified suggested that GTS can standardize plasmid construction workflow for the researchers working in metabolic engineering and other biotechnology research fields.

## Methods

**Chemicals**. All the chemicals were purchased from Sigma-Aldrich unless otherwise stated. All DNA oligos used in this work were synthesized by Integrated DNA Technologies and Guangzhou IGE Biotechnology LTD, and the DNA oligo sequence information is provided in Supplementary Data 1, 3, 4, 5, 6, and 7.

**Protocol**. We have prepared a detailed step-by-step protocol and uploaded it to Protocol Exchange[26].

**PCR**. PCR reaction solution in this work contained 1–5 µL of template DNA, 0.3 µL of 100 µM forward oligo, 0.3 µL of 100 µM reverse oligo, 25 µL of Q5® Hot Start High-Fidelity 2X Master Mix (M0494, New England Biolabs [NEB]), and ultrapure water to top up the volume to 50 µL. The cycling condition was based on the manufacturer's instruction. All amplified DNA fragments were separated by standard gel electrophoresis and then purified by using commercial column according to the manufacturer's instructions (GeneJET Gel Extraction Kit, K0691, Thermo Fisher Scientific). At the end of the purification, DNA was eluted from the column by using 40 µL of nuclease-free water (BUF-1180, 1st BASE Biochemicals [1st BASE]) in 1.7 mL Eppendorf tube.

**Iodine-based chemical reaction**. To cleave PS bond in DNA molecules (Fig. 1c), forty microliters of purified DNA solution was mixed with 5.5 μL of 1 M Tris solution (3021, 1st BASE, pH adjusted to 9) and 10 μL of 30 g L$^{-1}$ iodine solution (iodine: 207772, Sigma-Aldrich; solvent: ethanol), and was incubated at 70 °C for 5 min in a water bath. The solution was diluted with 250 μL of nuclease-free water, and purified by using commercial DNA purification column as described above.

**Preparation of DNA fragment**. Fragments can be amplified from various sources (e.g., plasmid, synthetic DNA, genomic DNA) by using PS-modified Foligos (Fig. 1c and Supplementary Fig. 6a). Iodine-based cleavage of the PS bond left two 1-nt SEs (C or T) on each fragment. The fragments treated by iodine solution must be purified by using column before being used in downstream applications (GeneJET Gel Extraction Kit, K0691, Thermo Fisher Scientific).

Fragments also can be amplified by using non-modified oligos containing selected Type IIS RE RS (Fig. 4a). The amplified fragment will be used in a one-pot RE-digestion and ligation procedure as shown in the below section of "One-pot enzymatic cleavage of fragment and barcoding".

Short fragments can be directly created by annealing Noligos (Supplementary Fig. 4a). The annealing reaction contained 50 μL of 100 μM G-Noligo and 50 μL of 100 μM A-Noligo. The annealing was done by using the following program in a thermo cycler: 98 °C for 2 min, 98 to 75 °C at rate of 0.1 °C/s, 75 °C for 2 min, 75 to 45 °C at rate of 0.1 °C/s, 45 °C for 2 min, 45 °C to 4 °C at rate of 0.1 °C/s and hold at 4 °C. The fragments prepared by annealing Noligos were diluted with nuclease-free water to 10–20 ng/μL for barcoding, and did not require purification. All the oligos used in this study are listed in Supplementary Data 1.

We have developed a Matlab App (GTS Oligo Designer) to help researchers design Foligos (PS-modified and non-modified), Noligos and Boligos. Step-by-step instruction on how to use the App is provided in this protocol[26]. The App can be downloaded from the link, https://www.mathworks.com/matlabcentral/fileexchange/71880-oligo-designer-for-gt-standard. If users do not have Matalb installed, an exe file (https://github.com/KangZhouGroupNUS/GTS-Oligo-Designer.exe.git) can be used to install the App.

**Phosphorylation and folding of Boligos**. Boligos need to have phosphate group at 5′ end and properly folded. To reduce oligo synthesis cost, we ordered regular oligos and used T4 kinase to add phosphate group. The phosphorylation reaction solution contained 1 μL of 100 μM Boligo, 2 μL of 10X T4 ligase buffer (B0202, NEB), 0.5 μL of T4 kinase (B0201, NEB) and 16.5 μL of nuclease-free water. Phosphorylation and folding of Boligo were done by using the following condition in a thermo cycler: 37 °C for 30 min (phosphorylation), 65 °C for 20 min (inactivation of T4 kinase), 98 °C for 2 min (DNA denaturing), 98–45 °C at rate of 0.1 °C/s, 45 °C for 2 min, 45 °C to 4 °C at rate of 0.1 °C/s, and hold at 4 °C. The prepared Boligos (diluted 4-fold by using ultrapure water) can be used in subsequent reactions without purification. All the Boligos (conventional and novel oligo design) used in this study are listed in Supplementary Data 3. The workflow for creating conventional Boligos is elaborated in Supplementary Fig. 2.

**Barcoding**. Prepared fragments with 1-nt SEs were ligated with Boligos by using a commercial kit (Blunt/TA Ligase Master Mix, M0367, New England Biolabs). The type of ligase was critical in this step (the results of using different ligases are shown in Supplementary Fig. 17). The ligation reaction solution contained 3 μL of fragment, 0.3 μL of 1.25 μM L(G/A)-Boligo, 0.3 μL of 1.25 μM R(G/A)-Boligo, and 3.6 μL of Blunt/TA Ligase Master Mix. The ligation reaction was done the following program in a thermo cycler: 25 °C for 5 min, and hold at 4 °C. It is critical to have sufficient amount of high quality fragment in the ligation reaction. The recommended minimal fragment concentration is 10 ng/μL for fragment no longer than 1 kb. The concentration refers to that of fragment with 1-nt SEs. If fragment is larger than 1 kb, we recommend to use at least 100 ng/μL; if fragment is larger than 2 kb, we recommend to use at least 200 ng/μL. We used Vacufuge (Eppendorf™ Vacufuge™ Concentrator) to concentrate fragment solution when its concentration was too low. Absorption spectrum (200–300 nm) of each fragment solution should be examined by using Nanodrop (NanoDrop™ 2000/2000c Spectrophotometers, Thermo Fisher Scientific) or a similar device to ensure there is a peak at 260 nm before its fragment concentration can be used in the calculation.

After ligation, corresponding Aoligos (Fig. 5c) were used to amplify correctly barcoded fragments by using PCR. This step was termed as ligation PCR. The ligation product was directly used as template DNA in PCR without purification/dilution. PCR product was purified and cleaved by Iodine-based chemical reaction if PS-modified Aoligos were used. Aoligos with or without PS-modification used in this study are listed in Supplementary Data 4. All barcoded fragments used in this study are listed in Supplementary Data 8.

**One-pot enzymatic cleavage of fragment and barcoding**. Four out of the five selected Type IIS REs were tested (Fig. 4b). The RE-digestion and ligation solution contained 1 μL of Type IIS RE (MboII [R0148S], HphI [R0158S], BmrI [R0600S] or BciVI [R0596S] NEB), 0.3 μL of 1.25 μM L(G/A)-Boligo, 0.3 μL of 1.25 μM R(G/A)-Boligo, 3 μL of fragment (the concentration of fragment is recommended in section "Barcoding"), and 4.6 μL of Blunt/TA Ligase Master Mix. The RE-digestion mixture was incubated at 37 °C for 1 h, and two microliters of the solution was used as

template in PCR to amplify barcoded fragment by using corresponding PS-modified Aoligos. PCR product was purified and treated by using iodine solution as specified in section "PCR" and "Iodine-based chemical reaction". All the non-modified oligos used to amplified the fragments with the 1-nt SEs are listed in Supplementary Data 1.

**Direct amplification of barcoded fragment**. Barcoded fragments can also be directly amplified from the constructed plasmid by using Aoligos from a plasmid if this plasmid contains the barcoded fragment (Figs. 6e–g and 7a, Supplementary Figs. 6a and 11a). In such case, barcoding was not required. The PCR products can be cleaved by Iodine-based chemical reaction, purified and used in DNA assembly.

**DNA assembly**. We used various DNA assembly methods in multi-tier workflows (Supplementary Fig. 16).

CLIVA[11]: The assembly reaction solution contained 1 μL of 10 mM MgCl$_2$, and 4 μL of mixture containing the equimolar barcoded fragments. The recommended minimal molar concentration of barcoded fragment is 10 nM. The following procedure was used: the mixture was heated by using a thermo cycler at 80 °C for 1 min, cooled down to 68 °C at a default speed of thermal cycle, kept for 10 min and then cooled down to 4 °C at 0.1 °C/s. The ligation mixture can be used for transformation directly.

Gibson method: The assembly reaction solution contained 4 μL of 2X Gibson Assembly Master Mix (E2611S, NEB), and 4 μL of fragment mixture containing the equimolar barcoded fragments. The ligation reaction was done at 50 °C for 15 min by using a thermo cycler. The ligation mixture can be used for transformation directly.

In-fusion cloning: The assembly reaction solution contained 1 μL of 5X In-Fusion HD Enzyme Premix (639645, Clontech), and 4 μL of fragment mixture containing the equimolar barcoded fragments. The ligation reaction was done at 50 °C for 15 min by using a thermo cycler. The ligation mixture can be used for transformation directly.

Golden Gate assembly: Three first-tier plasmids were constructed under GTS so that the barcodes incorporated with BsaI's RS and spacer can be used for creating three 4-nt SEs in the second-tier Golden Gate-based assembly (Supplementary Fig 10a and b). The enzymatic digestion solution contained 1 μL of BsaI (R3733S, NEB), a certain quantity of plasmid (~1 μg), 5 μL of 10X CutSmart® Buffer, ultrapure water to top up the reaction mixture to 50 μL. The RE-digestion mixture solutions were incubated at 37 °C for 1 h, and the targeted fragments were separated by using gel electrophoresis. The ligation solution contained 0.5 μL of T4 ligase (M0202L, NEB), 0.5 μL of 10X T4 ligase buffer, 4 μL of mixture containing the equimolar enzymatically digested fragment and backbone. The ligation reaction was done at 16 °C for 12 h by using a thermo cycler.

RE-based method: The first-tier plasmids were first constructed, and sequenced to ensure the barcode regions contained correct RE recognition sites. RE-digestion solution used in Supplementary Fig. 11 contained 1 μL of each RE (NotI [R3189S, NEB] and I-CeuI [R0699S, NEB]), a certain quantity of plasmids (~1 μg), 5 μL of 10X CutSmart® Buffer (B7204S, NEB), and ultrapure water to top up the reaction mixture to 50 μL. The RE-digestion mixture was incubated at 37 °C for 1 h, and the targeted fragments were separated by using gel electrophoresis. The ligation solution contained 0.5 μL of T4 ligase, 0.5 μL of 10X T4 ligase buffer, 4 μL of mixture containing the enzymatically digested barcoded fragment and backbone fragment (the molar ratio of insert to backbone was 3:1). The ligation reaction was done at 16 °C for 12 h by using a thermo cycler. RE-digestion and ligation used in Supplementary Fig. 12 were similar to the above except that BamHI (R0136S, NEB) and AatII (R0117S, NEB) were used.

*E. coli* transformation: One microliter of assembly solution was mixed with 17 μL of *E. coli* Dh5α heat-shock competent cell (C2987H, NEB) in a pre-chilled 1.7 mL tube on ice (Axygen). The tube was heat-shocked in a 42 °C water bath for exactly 35 s and was quenched on ice for 2 min The cell solution was mixed with 150 μL of SOC medium (B9020S, NEB) and directly plated on LB Agar plate that contained a proper antibiotic. The plate was incubated at the temperature required by specific applications. Usually colony appeared after 12–16 h when incubated at 37 °C.

Colony PCR and Sanger sequencing were carried out to determine assembly efficiency and sequencing accuracy of DNA assembly. The assembly accuracy was product of assembly efficiency and sequencing accuracy. Assembly efficiency was the ratio of the number of positive colonies (determined by colony PCR) to that of all the tested colonies. For each DNA assembly, one or more positive colonie(s) were cultured in LB with proper antibiotics overnight and the plasmids extracted from them were further tested by Sanger sequencing (Service provider: Axil Scientific, AITbiotech, and BioBasic). Sequencing accuracy was the ratio of the number of positive plasmids (free of mutation/deletion/insertion in sequenced region) to that of all the sequenced plasmids. The raw data used for statistical analysis of accuracy of plasmid construction (Fig. 1d) are provided in Supplementary Data 9. *E. coli* colony PCR reaction solution contained 1 μL of colony suspension (one single colony was resuspended in 100 μL of ultrapure water), 0.15 μL of 100 μM forward oligo, 0.15 μL of 100 μM reverse oligo, 5 μL of Q5 High-Fidelity 2X Master Mix, and 3.7 μL of ultrapure water. The oligo used in colony PCR for testing various DNA assembly methods are listed in Supplementary Data 7.

Yeast *in vivo* assembly[24]: The three first-tier plasmids (Supplementary Fig. 14a) were digested enzymatically, and the liberated fragments were assembled into a yeast plasmid (Supplementary Fig. 14d). The RE-digestion solution contained 1 μL

of each RE (NotI [R3189S, NEB] and/or SrfI [R0629S, NEB]), a certain quantity of plasmid (~1 μg, PGTP1 or PGTP2), 5 μL of 10XCutSmart® Buffer, and ultrapure water to top up the reaction mixture to 50 μL. The RE-digestion mixture solutions were incubated at 37 °C for 1 h, and the targeted fragments were separated by using gel electrophoresis. The obtained purified fragments (25 μL) were mixed and concentrated by using Vacufuge to around ~5 μL. The concentrated mixture was mixed with 50 μL of yeast competent cell (S. cerevisiae BY4741 competent cell was prepared by using S.c. EasyComp™ Transformation Kit [K505001, Thermo Fisher Scientific]). The Yeast transformation was done according to manufacturer's instruction, and the transformed cells were cultured on CSM-URA agar at 30 °C for 36–48 h. The plate with colonies was then visualized under Dark Reader (HL34T Hand Lamp, Clare Chemical) to verify their fluorescence. Five fluorescent and non-fluorescent colonies were resuspended in 100 μL ultrapure water individually. To lyse the cells, ten microliters of each colony resuspension was mixed with 10 μL of NaOH (40 mM), and incubated at 98 °C by using a thermo cycler for 30 min One microliter of the cell lysis solution was used as template for diagnostic PCR. The oligos used in yeast colony PCR are listed in Supplementary Data 7.

**Genome editing of E. coli.** For genome editing of E. coli MG1655_ΔrecA_-ΔendA_DE3, we utilized a reported two-plasmid CRISPR-Cas9 system[27]. The pTarget plasmids used in this study were listed in Supplementary Data 5. Colony PCR was performed to evaluate the efficiency of gene deletion at the selected loci. A full list of the targeted loci and oligos used in colony PCR are provided in Supplementary Data 5.

**Plasmids and strains.** A full list of plasmids and strains used and constructed in this study are listed in Supplementary Data 10. All the GenBank files of plasmids constructed in this study have been provided in Source data. Each E. coli and S. cerevisiae strain was derived from its parental strain through plasmid transformation. The standard electroporation protocol was used to transform plasmid into E. coli. Yeast transformation was done according to the instruction of kit that was used.

**Construction of combinatorial plasmid library.** To construct variants of M1 or M2 in the first-tier construction (Fig. 7a), each fragment (ppsA, aroE, tktA, aroG, tal or tyrA) was barcoded by three sets of RG-Boligo/LA-Boligo: RBS1/SRBS4, SRBS4/SRBS2, and SRBS2/3UTR1. The barcoded fragments can be assembled into the 12 plasmids when they were properly combined (Supplementary Fig. 9a). Each plasmid was verified by colony PCR and Sanger sequencing (Supplementary Data 9, P294 to P305). Each M1 plasmid (~1 ng/uL) was used as template to amplify one M1 fragment (Supplementary Fig. 9a) by using RG-Aoligo (5UTR) and LA-Aoligo (3UTR2). Each M2 plasmid (~1 ng/uL) was used as template to amplify one M2 fragment (Supplementary Fig. 9a) by using RG-Aoligo (3UTR2) and LA-Aoligo (N36). These M1 and M2 fragments were considered to be barcoded because Aoligos were used in the PCR, and thus can be directly assembled following our workflow (Fig. 1c). Six M1 fragments and six M2 fragments were equimolarly assembled with one plasmid backbone (barcoded) to create a mixture of 36 plasmids (a plasmid library). Technical details in this step: The iodine treated fragments were mixed, and 2 μL of the solution was mixed with 34 μL of E. coli Dh5α heat-shock competent cell, which after transformation were spread on 90 mm LB agar with a proper antibiotic; all the obtained colonies were resuspended in 6 mL of LB medium, and the mixed plasmids were extracted directly from this suspension. Two plasmid libraries were prepared, and each library had a different plasmid backbone (pSC101 + Amp^R or pAC + Spec^R).

The quality of each plasmid library was checked by using colony PCR. In the above step, colonies were randomly picked after competent cells were transformed with the mixture. Each colony was tested by using two pairs of oligos. The first pair targeted RO and ppsA (M1), and it would generate amplicons with varied lengths when the colonies contained plasmids that had ppsA at different positions of the operon (Supplementary Fig. 9b). Similarly, the second pair targeted tal (M2) and AR. The expected amplicon's length is listed in Supplementary Fig. 9c. Six colonies were tested for each plasmid library, and colony PCR results showed that each plasmid library indeed contained various plasmids (Supplementary Fig. 9b). The oligos used for colony PCR verification are listed in Supplementary Data 6.

Two microliters of plasmid mixture from each library was used to transform E. coli MG1655 ΔrecA_ΔendA_ΔpheA_ΔtyrR_DE3 through the standard electroporation procedure. Seventy-two colonies were randomly picked for each library, and each colony was screened to determine its ability of producing p-coumaric acid. After the screening, the top two p-coumaric acid-producing strains of each library were selected and the plasmids they harbor were sequenced to elucidate the responsible arrangement of the genes.

**Bacterium culture and metabolites quantification.** Each of plasmid TPP1-16 was used to transform E. coli TPS0 (genotype: MG1655 ΔrecA_ΔendA_ΔpheA_ΔtyrR_DE3) by using standard electroporation protocol. The resulting strains were named as TPS1-16. To test these strains, single colony was inoculated into LB with 50 μg mL$^{-1}$ of spectinomycin, and cultured at 37 °C/250 rpm overnight. One hundred microliters of the overnight grown cell suspension were inoculated into 10 mL of K3 medium (composition specified below) with 50 μg mL$^{-1}$ of

spectinomycin, and the culture was incubated at 30 °C/250 rpm until cell density reached 0.5–1.0 (OD600), at which the culture was induced by 0.1 mM of isopropyl β-D-1-thiogalactopyranoside (IPTG). One milliliter of the induced cells was transferred to a 14 mL round-bottom Falcon tube. If PthrC3 was used, IPTG induction was skipped. The cell culture was started in the 14 mL tube. The tube was incubated at 30 °C/250 rpm for 84 h.

At the end of incubation, one hundred microliters of 6 M HCl was added to 1 mL of cell culture broth for dissolving tyrosine crystals. The mixture was incubated at 37 °C/250 rpm for 30 min, and then centrifuged at 13,500×g for 5 min The supernatant was filtered using 13 mm, 0.2 μm Nylon filter.

To measure tyrosine titer, two microliters of the filtered supernatant prepared according to the above protocol was analyzed by high-performance liquid chromatography (HPLC, Shimadzu LC-10). The HPLC conditions are as follows: the column was Agilent ZORBAX Eclipse Plus C18 100 mm, an isocratic flow was used (the flow rate was 0.7 mL/min and the mobile phase consists of 10% [v/v] acetonitrile and 90% [v/v] aqueous solution containing 0.1% [v/v] trifluoroacetic acid), the column temperature was 30 °C, and the detector was UV detector (wavelength: 254 nm).

To measure p-coumaric acid titer, three hundred microliters of acidified medium (without centrifugation) was mixed with 700 μL of acetonitrile, the mixture was incubated at 30 °C for 1 h, the mixture was centrifuged at 13,500×g for 5 min, and two microliters of the supernatant was analyzed by HPLC (Shimadzu LC-10). The HPLC conditions are as follows: the column was Agilent ZORBAX Eclipse Plus C18 100 mm, an isocratic flow was used (the flow rate was 1 mL/min and the mobile phase consists of 35% [v/v] acetonitrile and 65% [v/v] aqueous solution containing 0.1% [v/v] trifluoroacetic acid), the column temperature was 30 °C, and the detector was UV detector (wavelength: 285 nm).

K3 medium consisted of 89.8% (v/v) of K3 basal medium, 10% (v/v) carbon source stock solution and 0.17% (v/v) K3 master mix. K3 basal medium was prepared by dissolving 4 g of (NH$_4$)$_2$HPO$_4$, 13.3 g of KH$_2$PO$_4$, 60 mg of L-phenylalanine in 1 L of deionized water, adjusting pH to 7 by using 6 M NaOH, and autoclaving the medium. The carbon source stock solution was 200 g L$^{-1}$ glucose solution (autoclaved). K3 master mix was prepared by mixing 2.5 mL of 0.1 M ferric citrate solution (autoclaved), 1 mL of 4.5 g L$^{-1}$ thiamine solution (filtrated through 0.2 μm filter), 3 mL of 4 mM Na$_2$MoO$_3$ (autoclaved), 1 mL of 1000X K3 trace elements stock solution (autoclaved) and 1 mL of 1 M MgSO$_4$ solution (autoclaved). We prepared 1000x K3 trace elements stock solution by dissolving 5 g of CaCl$_2$·2H$_2$O, 1.6 g of MnCl$_2$·4H$_2$O, 0.38 g of CuCl$_2$·4H$_2$O, 0.5 g of CoCl$_2$·2H$_2$O, 0.94 g of ZnCl$_2$, 0.0311 g of H$_3$BO$_3$ and 0.4 g of Na$_2$EDTA·2H$_2$O in 1 L of deionized water, and autoclaved this solution.

**Culture and analysis of GFP-expressing E. coli.** Each of plasmid A0-A8 was used to transform E. coli BL21 (DE3) (C2527H, NEB). Single colony was inoculated into LB with 50 μg mL$^{-1}$ of spectinomycin and cultured at 37 °C/250 rpm overnight. Fifty microliters of the overnight grown cell suspension was inoculated into 5 mL of K3 medium with 50 μg mL$^{-1}$ of spectinomycin, and the culture was incubated in 50 mL Falcon tube at 37 °C/250 rpm for 24 h. Optical density 600 (OD600) of cell suspensions was determined by a microplate reader (Varioskan LUX Multimode Microplate Reader, Thermo Fisher Scientific). For each sample, two hundred microliters of cell suspension were loaded into a well of 96-well optical plate and assayed with the following parameter setting: excitation wavelength was 483 nm, emission wavelength was 535 nm, measurement time was 100 ms, and the bandwidth of excitation and emission light were 12 nm. Fluorescence signal was normalized by OD600 of cell suspension to calculate specific fluorescence signal.

**Reporting summary.** Further information on research design is available in the Nature Research Reporting Summary linked to this article.

## Data availability
All the data supporting the findings of this study are available within the paper and the supplementary information file. Source data of gel analysis for Fig. 2a and b, Fig. 4c are provided with the paper. Source data of Genbank files of plasmids constructed in this study are provided with the paper. All other relevant data are available from the authors upon reasonable request.

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

## Acknowledgements

This work was supported by National University of Singapore Start-up Grant (R-279-000-452-133), Singapore Ministry of Education Tier 1 Grants (R-279-000-478-112 and R-279-000-494-114), Singapore National Research Foundation CRP Grant (R-279-000-512-281), and Disruptive & Sustainable Technologies for Agricultural Precision Grant (R-279-000-531-592). We acknowledge valuable discussions, suggestions and proof-reading of manuscript from W. Wang. We acknowledge useful guidance from L. Yang on using instruments in the Department of Chemical & Biomolecular Engineering, National University of Singapore. We thank S. Somasundaram, S. Panda, Q. Pan and J. Zhen for providing information of the plasmids they constructed by using GTS.

## Author contributions

X.M. and K.Z. conceived the project. X.M. and K.Z. designed the experiments, analyzed the results and wrote the manuscript. X.M. performed all the experiments with the assistance from other co-authors. H.L., X.C., Y.L. and H.L. assisted X.M. with screening of plasmid library. H.L., X.C., W.N, B.H. and P.N. assisted X.M. with construction of the plasmids for editing *E. coli* genome and expressing GFP in yeast.

## Additional information

**Competing interests:** An international patent application has been submitted on the technologies reported in this article: Zhou, K., Ma, X. & Liang, H. Ligation and/or assembly of nucleic acid molecules. Application No. PCT/SG2018/050528 (National University of Singapore, Singapore). The remaining authors declare no competing interests.

