## [Peer Review File · Nature Communications]

Reviewers' Comments:

Reviewer #1:

Remarks to the Author:

In this manuscript, Ma et al. described a standard DNA assembly approach to assemble multiple pieces of DNA together to form a circular plasmid. In their approach, DNA fragments are first attached to sequences called barcodes at both ends using DNA ligation. After this step, the fragment-barcode sequence is again amplified using primers containing phosphorothioate bonds and after amplification and chemical cleavage of phosphorothioate bonds, the fragments can be assembled together with a modified CLIVA approach. The authors claimed that the advantage of this approach compared to other DNA assembly methods is the fact that oligonucleotides used for barcodes and barcode amplification are reusable and could lower the cost of DNA assembly. Although this approach, if adapted by a whole community, could potentially lower the cost of oligonucleotides used for DNA assembly, it does not significantly lower the cost and time of the whole DNA assembly process. In addition, there are a few other issues that the authors need to address in order to improve the quality of the manuscript. Overall, the authors made many claims that were not supported by their results and the novelty of the work was not significant enough for a high-impact journal like Nature Communications.

Major comments:

- 1) The limits for number of DNA fragments and final plasmid size should be clearly stated in the abstract. The authors mentioned that they were able to assemble up to 14 fragments into one plasmid, but I was not able to find the example in the manuscript. The authors also claimed that they can standardize construction of almost any plasmid, which is not valid.
- 2) The authors need more convincing evidence to support their claim of why this method is superior to comparable methods like Gibson assembly or CLIVA itself. The time-frame of this approach is clearly longer than Gibson assembly. GTas also has multiple steps of amplification and purification. It also uses oligos with phosphorothioate bonds which are more expensive and also not available in lowest scale of DNA synthesis.
- 3) One of the important questions is that whether all barcodes are actually compatible with each other or form an efficient set. For example, if seven fragments are assembled to form a 12 kb plasmid with one set of barcodes, can this assembly be performed with a different set of barcodes? Examples of this could show robustness of this approach and support their claims.

Minor comments:

- 1) In comparison of eCLIVA to CLIVA, the authors' claim that the assembly efficiency was significantly improved is not valid unless they performed the same assembly using both approaches. Creation of a 22 kb plasmid is significantly harder than a 4.2 kb plasmid.
- 2) Supplementary Figure 10 is not very clear for comparison of different approaches. I could not figure out the overall time needed for each assembly method by the pie chart.
- 3) In the flexible editing of constructed plasmids, it is mentioned that GTas is able to replace or remove fragments from any plasmids assembled under GTas. The authors' approach for performing editing on a plasmid is however PCR-based., which limits its applications. For example, removing a 1 kb fragment from a 17 kb plasmid is almost impossible using this approach.

Reviewer #2:

Remarks to the Author:

General:

Ma et al. report a new DNA cloning method dubbed GT assembly to facilitate the assembly of

multiple DNA fragments for building entire plasmids or various sizes. As a key feature the method implements barcodes to connect neighboring DNA fragments. The barcodes can, for example, be RBSs or just short linker sequences. A novelty of this work is that the cloning fragments are PCR-amplified using oligos that have a phosphorothioate (PS) bond after the first (or few) nucleotide(s) at the 5' end; chemical cleavage then makes it possible to generate overhang ends. The PS bond cleavage allows generating fragments with just one nucleotide, which is the minimal length of any conserved sequence.

Ligating DNA fragments with only 1-nt sticky ends is typically inefficient. Authors solved the problem of ligating such fragments with the barcodes: they sealed one end of each barcode half; each barcode is represented by a single oligonucleotide that has a stem-loop structure at one end (this oligo was dubbed boligo, for barcoding oligo). Ligation of two such sealed barcode halves to one fragment creates a circular DNA molecule to which no further DNA parts can ligate, unless the stem-loop structure is removed. This is achieved by a simple chemical reaction (that acts on a PS bond introduced by a subsequent PCR with primers, called Aoligos, containing this PS bond). As the PS bond is located centrally in the Aoligos, chemical cleavage leads to longer (15-20 nt) overhangs that allow annealing of different barcoded fragments at elevated temperature (45°C, using Taq ligase).

Authors used GT assembly to generate 370 different plasmids in total.

The possibility to generate plasmid libraries for the combinatorial optimization of pathways is an important aspect of GT assembly with likely useful applications in many metabolic engineering projects in *E. coli*.

I am not so much convinced that the GT assembly method will indeed lead to more efficient utilization of DNA synthesis power (in particular considering the still dropping prices for oligo and gene syntheses). To me, authors are overstating the potential of reusing DNA fragments or oligos too much. However, there is the potential that other researchers adopt the newly developed technology in their own synthetic biology projects.

Limitation of the method:

If I see it correctly, GT assembly can only be employed in protocols that aim at generating entire plasmids from fragments (about 2 – 7 fragments seem to reliably work, see lines 170/171, and Figure 1d), and not to insert (several) fragments into a given plasmid. This is most likely not optimal for most researchers as many of them prefer to only reconstruct the part of the plasmid they indeed need to change. The longer fragments that form the basis of the plasmid need to be amplified from a template (if not produced by de novo gene synthesis); here, the potential limitation of GT assembly is that PCR can introduce sequence errors into longer fragments. Also, the amplification of the barcoded fragments with Aoligos occurs via PCR. This is something important to consider; currently, authors have not addressed this issue of GT assembly appropriately in their manuscript. In principle, each newly synthesized plasmid should best be sequenced in its entirety. Is this, what authors are recommending?

Note, the accuracy of plasmid construction is 86% based on Sanger sequencing (lines 168/169); however, here it is not clear whether the errors that occur mostly come from PCR-introduced nucleotide exchanges, or from an erroneous annealing of fragments and barcodes (in wrong order or numbers). Authors should comment on it.

Is GT assembly suitable for pathway engineering in others hosts?

Authors developed GT assembly for pathway engineering in *E. coli*. In principle, however, the method can also be used for building plasmids for other bacterial but also eukaryotic hosts, e.g. *Saccharomyces cerevisiae* (although in yeast, plasmid-based pathway engineering is often suboptimal compared to the engineering of pathways using genome-integrated genes). Authors should shortly discuss the option of extending their method to other organisms.

With respect to *S. cerevisiae*, another very prominent host in synthetic biology projects, please

note that it is an excellent organism for performing high-throughput in vivo cloning at very low cost. The obvious question, therefore, is: would GT assembly be able to compete in yeast with such in vivo cloning methods? Authors should discuss this.

Other comments:

- Abstract, line 26: authors mention that the method can be used to assemble up to 14 DNA parts in one round of operation. Where is this described in the main text? Actually, authors speak about 2 – 7 fragments in the main text, see also Fig. 1d.
- Line 51: I think it should read ‘but has NOT yet been adopted widely’.
- Line 52: instead of speaking about ‘flaws’ I recommend to speak about ‘limitations’.
- Lines 82-84: authors say that ‘most functional DNA sequences can be defined as fragments and/or barcode’. What does this mean with respect to the re-use of fragments (or the offer of standardized fragments by a company, in a kit)? For example, if one lab wants to have a functional DNA part be represented by a ‘fragment’, and another by a ‘barcode’, the reusability idea would be limited to some extent, or? Authors need to comment on this.
- Line 85: the term ‘complementary’ in DNA terms typically means complementarity of DNA sequences (or forward and reverse strands of DNA). Here, however, authors mean the complementing other half of the barcode. Therefore, I am suggesting to use the term ‘complementing’.
- Line 132: Fig. 3b is mentioned before Fig. 3a (line 180).
- Lines 150/151: authors state that PS bonds were introduced into the fragments AFTER PCR; however, in line 149 they say that the PCR oligonucleotides had PS bonds, meaning that the PCR bonds are introduced into the fragments during the PCR amplification. This needs clarification.
- Line 151: ‘simple chemical reaction’; which one? Although it is outlined in the methods part of the manuscript, the reader would benefit from getting a short note here as well.
- Lines 163/164: The term ‘Aoligo’ is introduced here by referring to Fig. 3c, while already Fig. 2a,b mentions Aoligos. If possible, authors should introduce the term earlier in the manuscript.
- Lines 170/171: typically, authors assembled 2 – 7 fragments for plasmid generation, with an accuracy of 86%. Cloning accuracy was similar with different numbers of fragments, and different plasmid sizes over quite a considerable length range (2.4 to 13 kb). However, authors should indicate the lengths of the fragments that were assembled in these experiments, and refer to the respective supplementary table where this is reported in more detail.
- Lines 188 ff: where is the information about the replication origins (RO) and antibiotic resistance markers (AR) given?
- Line 207: the 16 plasmids constructed for engineering E. coli to overproduce tyrosine are not included in Fig. 1d, it seems (as only 8 plasmids are represented there for 6-fragment assemblies). Why not?
- Lines 286 – 290: This chapter is not very informative. Authors should better explain what was done; a reference to two supplementary figures is in my opinion not enough.
- Line 631: Is reference 1 correct for the CLIVA method? Note, that in the supplementary data file authors refer to a different reference for CLIVA.
- I have two questions to Fig. 1d: (FIRST) Authors state they have successfully constructed 370 plasmids using GT assembly (main text, line 166), but in Fig. 1d only about 335 plasmids are represented. Seems, more than 30 plasmids are not shown in the figure. What is the reason for it? (SECOND) The largest plasmids, indicated by the yellow and red color codes, were assembled from 5 fragments, and the medium size plasmids (green) mostly from 4 fragments, while 6- and 7-fragment assemblies were only used for smaller plasmids. Does this mean, that constructing larger plasmids with 6 or 7 fragments does not work? Or did authors simply not test it? Authors should comment on it.
- Figure 4: several panels mention plasmids TPP0, TPP18 and PCAP1, however, those are not mentioned in the main text and not explained in the figure legend. Please clarify what these plasmids are/why they were included.

Response to Reviewers' Comments:

We sincerely appreciate the reviewers' valuable and constructive suggestions. Based on the reviewers' comments, we have done experiments to address the limitations of our original design. These experiments have led to substantial improvements to GT standard (GTS), and have provided data that in our opinion can address all the concerns of the reviewers. Please find the point-to-point response to reviewers' comments below. [Comments from reviewers are highlighted in golden, and responses are highlighted in blue]

Reviewer #1 (Remarks to the Author):

In this manuscript, Ma et al. described a standard DNA assembly approach to assemble multiple pieces of DNA together to form a circular plasmid. In their approach, DNA fragments are first attached to sequences called barcodes at both ends using DNA ligation. After this step, the fragment-barcode sequence is again amplified using primers containing phosphorothioate bonds and after amplification and chemical cleavage of phosphorothioate bonds, the fragments can be assembled together with a modified CLIVA approach. The authors claimed that the advantage of this approach compared to other DNA assembly methods is the fact that oligonucleotides used for barcodes and barcode amplification are reusable and could lower the cost of DNA assembly. Although this approach, if adapted by a whole community, could potentially lower the cost of oligonucleotides used for DNA assembly, it does not significantly lower the cost and time of the whole DNA assembly process. In addition, there are a few other issues that the authors need to address in order to improve the quality of the manuscript. Overall, the authors made many claims that were not supported by their results and the novelty of the work was not significant enough for a high-impact journal like Nature Communications.

We thank the reviewers' comments. We totally agree that if researchers use GTS in their plasmid construction, the cost and time for plasmid construction can be reduced, since under GTS, there is no need to order customized oligos. In such case, the time of waiting the delivery of these customized oligos (at least one day) can be saved for the new plasmid construction. If this practice is adopted by a whole community, we would expect much more frequent reuse of standard oligos (Foligo/Aoligo/Boligo), which can help researchers save both time and cost of plasmid construction.

We need to clarify that GTS offers a near scarless, modular DNA assembly standard instead of a DNA assembly method. After receiving the comments, we have done experiments to prove that another two popular DNA assembly methods are also able to construct plasmid under GTS.

We sincerely apologize for the misunderstanding caused by some ambiguous statements in our original manuscript. We have clarified them below and in the revised manuscript. We have also provided additional demonstrations to address the limitations of our original design and to showcase the convenience it can bring to researchers who need to construct plasmids.

Major comments:

1) The limits for number of DNA fragments and final plasmid size should be clearly stated in the abstract. The authors mentioned that they were able to assemble up to 14 fragments into one plasmid, but I was not able to find the example in the manuscript. The authors also claimed that they can standardize construction of almost any plasmid, which is not valid.

The 14 DNA parts referred to 7 fragments and 7 barcodes. We used the number because (1) we did not define fragment and barcode in the abstract, and (2) many of the barcodes were indeed useful biological parts such as ribosomal binding sites. We are sorry for the confusion it has caused, and did not intentionally inflate what we had achieved. We have changed the statement to "assemble up to seven such barcoded fragments " in the revised abstract. The details of assembling 7 barcoded fragments can be found in **Figure 5a, 5b and 5d**, and **Supplementary figure 8**.

The claim that GTS can standardize construction of almost any plasmid has been rephrased. We now claim that GTS can standardize constructing plasmid by using various DNA assembly methods (**Supplementary figure 5**), and also can be used to modify existing plasmid by inserting single and composite DNA part by using restriction enzyme (RE)-based methods (**Supplementary figure 12**). We have experimentally demonstrated this claim by the newly developed two-tier workflows (**Supplementary figure 16**).

We revised the claim in the revised manuscript that GTS can be proceeded by using various DNA assembly methods (**line 154-162**, the line number refers to that in the manuscript without tracked changes).

2) The authors need more convincing evidence to support their claim of why this method is superior to comparable methods like Gibson assembly or CLIVA itself. The time-frame of this approach is clearly longer than Gibson assembly. GTas also has multiple steps of amplification and purification. It also uses oligos with PS bonds which are more expensive and also not available in lowest scale of DNA synthesis.

We wish to clarify that we are not providing a DNA assembly method that competes with Gibson assembly or CLIVA. Once a DNA fragment is barcoded, they can also, theoretically, be assembled by other overlapping sequence-based DNA assembly methods, such as CLIVA, Gibson, In-fusion, etc.

To experimentally prove the compatibility of GTS with popular DNA assembly methods, we tested three methods (CLIVA, Gibson and In-fusion cloning methods) to assemble five or seven fragments under GTS (**Supplementary figure 5**). All the methods worked well in all assembly tests under GTS (**Supplementary figure 5**).

As the reviewer mentioned, one limitation of GTAs was the use of expensive PS-modified oligos, which were required in both barcoding and assembly in the workflow we employed.

After receiving the reviewer's comment, we developed a new workflow to generate fragment with 1-nt sticky ends (SEs) by using non-modified oligos. We experimentally demonstrated that a panel of type IIS REs can accurately produce the 1-nt SEs. We further developed a one-pot reaction to create 1-nt SEs and use them to barcode fragments (**Figure 3a**). Barcoded fragments can be amplified in the second round of PCR, and further be assembled into a plasmid. We have tested this procedure by using them to construct three plasmids, and all of the plasmid construction were successful (**Supplementary figure 7**). We have found five REs that can be used to generate such 1-nt SEs. Any one of them can be used with a fragment if the fragment does not have an internal recognition sequence of these REs. We analyzed the existing 332 fragments in our lab, and found that 89.5 % of them can be barcoded by at least one of five REs. If one intends to use this method to barcode a fragment that contains recognition sequence of these REs, the internal cutting sites can be removed through silent mutagenesis, just as what is being done with Golden Gate assembly.

As for Aoligo, non-modified oligos can be used to replace PS-modified oligo when Gibson method or In-fusion cloning is used.

We have highlighted in the revised manuscript that we are developing a DNA assembly standard, instead of a DNA assembly method (**line 154-156** and **line 391-393**). We have described in the revised manuscript that Gibson assembly and In-fusion can be used to replace CLIVA (**line 154-156**). We have added a new section to **RESULTS** to describe the new method for generating the 1-nt SEs without using PS-modified oligos (**line 179**).

3) One of the important questions is that whether all barcodes are actually compatible with each other or form an efficient set. For example, if seven fragments are assembled to form a 12 kb plasmid with one set of barcodes, can this assembly be performed with a different set of barcodes? Examples of this could show robustness of this approach and support their claims.

After received the reviewer's comment, we did the experiments to prove that different sets of barcodes can be used in 7-fragment assembly.

Three 7-fragment assembly tests were done by using various sets of barcodes to construct similar plasmids to TPP7 (**Figure R1**). Two out of three worked. The one that failed could be due to a specific barcode that needed to be optimized because it worked after the barcode was replaced. Another argument we hope to use to prove that most barcodes are compatible with each other is that we did not consider compatibility before we attempted to assemble the 436 plasmids we have constructed and most of them worked.

We have added the discussions (**line 361-371**) in the revised manuscript.

Figure R1 Testing various sets of barcodes in 7-fragment assembly. **(a)** The barcode set used to construct the plasmid TPP7 for overproducing tyrosine. **(b)** Six barcodes in the plasmid TPP7 were replaced in the plasmid TPP7-test1 except RBS1, and seven barcodes in the plasmid TPP7 were replaced in the plasmid TPP7-test2 and TPP7-test3. Blue thick horizontal bars represent fragments; AR: Antibiotic resistance

marker (Spec^R); RO: Replication origin (pMB1); pLac: Lac promoter; *aroG*: a gene encoding mutated *E. coli* 3-deoxy-7-phosphoheptulonate synthase; *tyrA*: a gene encoding mutated *E. coli* fused chorismate mutase/prephenate dehydrogenase; t7t: T7 terminator; LacI: Lac promoter repressor expression cassette. Green texts indicate barcode; N21, N22, N23, N24, LK3, N32, N36, N37, 5UTR: non-functional connectors; RBS4: ribosomal binding site; pJ23119: a constitutive promoter; 3UTR2: 3' untranslated region. **(c)** Analysis of PCR products of the barcoded fragments that would be assembled to be the plasmids TPP7-test1, TPP7-test2 and TPP7-test3. **(d)** Transformation efficiencies of the three plasmids assembly. The PCR products of the barcoded fragments used to construct the plasmid TPP7 are shown in **Supplementary figure 8b**. The results of colony PCR used to verify the assembly efficiency of TPP7 **(e)**, TPP7-test1, TPP7-test2 and TPP7-test3 **(f)**. The correct amplicons were highlighted in white rectangle boxes. **(g)** Sequencing results of the plasmid extracted from the colonies verified by colony PCR. The results indicate that all the sequenced plasmids are correct. The sequencing was performed to cover the important regions (e.g., barcode region and coding sequence [CDS]).

Minor comments:

1) In comparison of eCLIVA to CLIVA, the authors' claim that the assembly efficiency was significantly improved is not valid unless they performed the same assembly using both approaches. Creation of a 22 kb plasmid is significantly harder than a 4.2 kb plasmid.

We apologize for this unfair comparison.

We have experimentally compared CLIVA and eCLIVA (**Supplementary figure 5**). Original CLIVA did have a good assembly efficiency and accuracy in the assembly of 5 and 7 fragments. When we developed eCLIVA, we hypothesized that adding Taq ligase could help the assembly efficiency, thus improving the transformation efficacy. It was found that eCLIVA (eCLIVA_0.33h_5 fragments) in 5-fragment assembly did not work better than the original CLIVA method in terms of assembly efficiency and accuracy (CLIVA_0.33h_5 fragments, **Supplementary figure 5d** and **5e**). With longer incubation time (eCLIVA_1h_5 fragments), the CFU/ μ g DNA was slightly increased in assembly of 5 fragments. In the 7-fragment assembly comparison, to our surprise, CLIVA method achieved better results than eCLIVA method (**Supplementary figure 5d**). Since mixed comparison results have been obtained, and thanks to the reviewer's comment, we have withdrawn our claim that eCLIVA method was superior to CLIVA method in the revised manuscript.

2) Supplementary Figure 10 is not very clear for comparison of different approaches. I could not figure out the overall time needed for each assembly method by the pie chart.

The comment made us to question if we should do the comparison and made us to realize that we should not, because we are developing a standard not an assembly method. As mentioned before, the standard can be used with a few popular assembly methods. We have thus removed this supplementary figure.

3) In the flexible editing of constructed plasmids, it is mentioned that GTAs is able to replace or remove fragments from any plasmids assembled under GTAs. The authors' approach for performing editing on a plasmid is however PCR-based, which limits its applications. For example, removing a 1 kb fragment from a 17 kb plasmid is almost impossible using this approach.

Indeed, it would be difficult to use PCR to modify the larger plasmid. This limitation can be solved if we include RE or homing endonuclease recognition sequence in some barcodes. The idea is that we use RE to reuse the majority of the backbone and only edit a smaller portion of the plasmid by using PCR-based method.

After receiving the reviewer's comment, we have developed a workflow that is able to remove one fragment from a 12.5 kb plasmid under GTS without using PCR to amplify the majority of the backbone. In this example, we intended to remove fragment *tal* from operon plasmid 1 (12.5 kb, **Figure R2**) to make this *p*-coumaric acid-producing plasmid to be a tyrosine-producing plasmid (**Figure R2**). We included RE sites in the barcodes that connected the operons. When we needed to remove fragment *tal*, we used two REs to remove the operon that contained *tal* and inserted a variant of the operon that did not contain *tal*. We then inserted a piece of DNA that carries the desired changes to the backbone through the same endonucleases and ligases. The short variant of the operon was constructed by using PCR-based method under GTS, but the other parts of plasmids were recycled and reused by using REs (**Figure R2**). The plasmid construction was done efficiently and the fragment was successfully removed, which was confirmed by Sanger sequencing.

We have added this workflow to the last section of **RESULTS** in the revised manuscript (**line 326-337**).

Figure R2 RE-based two-tier workflow under GTS. **(a)** Barcodes of 5UTR, 3UTR2 and N36 used in the second-tier assembly for plasmid library construction (**Figure 5a**) were replaced by 5UTR, REC1 and I-CeuI. In the first-tier, three plasmids carrying M1, M2 and M3 were constructed by using CLIVA, and the results of colony PCR and sequencing are provided on the top of plasmid schematic maps. After that, the plasmids carrying M1 and M2 were used as template for the second round of PCR to generate M1 and M2 fragments. The fragments were assembled with one plasmid backbone (RO+AR: pAC+Spec^R) to be operon plasmid 1 for pHCA production. **(b)** To convert operon plasmid 1 to be a tyrosine production plasmid, *tal* needs to be removed. M3 plasmid was constructed in the first-tier accordingly. **(c)** The sequences of REC1 barcode (containing three 8-bp RE recognition sites) and I-CeuI barcode (containing a long 19-bp of RE recognition site). **(d)** The REs included in REC1 and I-CeuI. **(e)** Through two REs digestion of operon plasmid 1 and M3 plasmid, the backbone (containing M1) and M3 fragment were obtained. The desired bands are indicated by black arrows in gel image. The purified M3 fragment was ligated with backbone containing M1 by using T4 ligase for 12 h at 16 °C. **(f)** The operon plasmid 2 was constructed by assembling M3 with backbone carrying M1. **(g)** The transformation efficiency of two-fragment ligation was evaluated by using CFU/ μ g DNA. **(h)** Colony PCR using the oligos targeting on the whole region of M3 region indicated that all six randomly picked colonies was positive. **(i)** Two plasmids extracted from two positive colonies were sequenced to be correct at the regions of three barcode (5UTR, REC1 and I-CeuI). Other regions in final plasmid that were covered in sequencing are free of undesired sequence errors.

Reviewer #2 (Remarks to the Author):

General: Ma et al. report a new DNA cloning method dubbed GT assembly to facilitate the assembly of multiple DNA fragments for building entire plasmids or various sizes. As a key feature the method implements barcodes to connect neighboring DNA fragments. The barcodes can, for example, be RBSs or just short linker sequences. A novelty of this work is that the cloning fragments are PCR-amplified using oligos that have a phosphorothioate (PS) bond after the first (or few) nucleotide(s) at the 5' end; chemical cleavage then makes it possible to generate overhang ends. The PS bond cleavage

allows generating fragments with just one nucleotide, which is the minimal length of any conserved sequence.

Ligating DNA fragments with only 1-nt sticky ends is typically inefficient. Authors solved the problem of ligating such fragments with the barcodes: they sealed one end of each barcode half; each barcode is represented by a single oligonucleotide that has a stem-loop structure at one end (this oligo was dubbed boligo, for barcoding oligo). Ligation of two such sealed barcode halves to one fragment creates a circular DNA molecule to which no further DNA parts can ligate, unless the stem-loop structure is removed. This is achieved by a simple chemical reaction (that acts on a PS bond introduced by a subsequent PCR with primers, called Aoligos, containing this PS bond). As the PS bond is located centrally in the Aoligos, chemical cleavage leads to longer (15-20 nt) overhangs that allow annealing of different barcoded fragments at elevated temperature (45°C, using Taq ligase).

Authors used GT assembly to generate 370 different plasmids in total.

We have updated the number of plasmid constructed under GTS to be 436 in the revised manuscript.

The possibility to generate plasmid libraries for the combinatorial optimization of pathways is an important aspect of GT assembly with likely useful applications in many metabolic engineering projects in *E. coli*.

I am not so much convinced that the GT assembly method will indeed lead to more efficient utilization of DNA synthesis power (in particular considering the still dropping prices for oligo and gene syntheses). To me, authors are overstating the potential of reusing DNA fragments or oligos too much. However, there is the potential that other researchers adopt the newly developed technology in their own synthetic biology projects.

In the revised manuscript, we have taken the advice from the reviewer to shift the focus of this study away from improving utilization of DNA synthesis power. We have further improved GTS in terms of being compatible with more DNA assembly methods and hope that it can serve more researchers in the biotechnology field.

Limitation of the method: If I see it correctly, GT assembly can only be employed in protocols that aim at generating entire plasmids from fragments (about 2 – 7 fragments seem to reliably work, see lines 170/171, and Figure 1d), and not to insert (several) fragments into a given plasmid. This is most likely not optimal for most researchers as

many of them prefer to only reconstruct the part of the plasmid they indeed need to change.

This comment has urged us to think how to use a given plasmid under GTS. The core of this new standard is the barcoding step, in which a pair of adapters can be added to a DNA fragment almost seamlessly. Since there is almost no restriction on what adapters can be added, we hypothesized that GTS should allow adding restriction enzyme (RE) sites or other sequences that are needed to edit a given plasmid.

To test this hypothesis, we tested editing pHT01, a commercial *Bacillus subtilis* expression plasmid, by inserting an operon into it under GTS. First, we constructed the first-tier plasmid carrying a three-gene operon flanking with two barcodes contained recognition site of BamHI and AatII, respectively (**Figure R3**). After enzymatic digestion of this plasmid and pHT01, the insertion of the operon to pHT01 was done by using a conventional ligation reaction. The plasmid was successfully constructed with high efficiency (**Figure R3**).

By incorporating specific RE sites into barcode, it is very flexible to select suitable RE for the insert (single or composite DNA fragments) from a large collection of RE-containing barcode. Through a set of GTS barcodes that contain various REs and RBSs, any combination of a few genes can be overexpressed as an operon by using any conventional expression vectors that use RE-based cloning.

We have added this feature of GTS to the last section of **RESULTS** in the revised manuscript (**line 339-354**).

Figure R3 Insert an operon constructed under GTS into a given plasmid. A two-tier workflow was established for introducing a three-gene operon (biosynthesizing naringenin from *p*-coumaric acid) into pHT01, a plasmid that can be used for expressing proteins in *Bacillus subtilis*. **(a)** In the first-tier, the plasmid (RECP1) was constructed by assembling four fragments (4cl, chi, chs and backbone [Spec^R+pAC]), and the results of colony PCR and sequencing are provided on the top of plasmid schematic maps. The RECP1 and pHT01 were enzymatically digested by two REs (BamHI/AatII). The liberated operon fragment was ligated with the pHT01B fragment through 4-bp SEs at 16 °C for 12 h by using T4 ligase. 4cl: 4-coumaroyl-CoA ligase; chi: chalcone isomerase; chs: chalcone synthase; Spec^R: antibiotic resistance marker; pAC: p15 replication origin; LacI: Lac promoter repressor expression cassette; Pgrac: grac promoter used in *B. subtilis*; TrpAt: terminator; Cm^R: antibiotic resistance marker used in *B. subtilis*; ColE1: replication origin used in *E. coli*; Amp^R: antibiotic resistance marker used in *E. coli*; RepA: replication origin used in *B. subtilis*. **(b)** The sequence of two barcodes (REC3 and REC4). **(c)** RE recognition sites included in REC3 and REC4. **(d)** Gel electrophoresis analysis of enzymatically digested products of RECP1 and pHT01, and the targeted bands are highlighted in white rectangle boxes. **(e)** The transformation efficiency of two-fragment ligation was evaluated by using CFU/μg DNA. **(f)** Colony PCR using the one primer (LacI-f) targeting on backbone region and one targeting on the operon region (4cl-r) indicated that 9 out of 10 randomly picked colonies was positive. **(g)** RE-digestion confirmed two plasmids extracted from two positive colonies have correct operon fragment. **(h)** Sequencing of these two plasmid confirmed that all the barcode regions were accurate. The other regions of the final plasmid that could be covered in sequencing were also free of sequencing errors.

The longer fragments that form the basis of the plasmid need to be amplified from a template (if not produced by de novo gene synthesis); here, the potential limitation of GT assembly is that PCR can introduce sequence errors into longer fragments. Also, the amplification of the barcoded fragments with Aoligos occurs via PCR. This is something important to consider; currently, authors have not addressed this issue of GT assembly appropriately in their manuscript. In principle, each newly synthesized plasmid should best be sequenced in its entirety. Is this, what authors are recommending?

Indeed, the undesired sequence errors can be introduced into plasmid during PCR. Also, the size of plasmid constructed could be limited because it could be difficult to amplify the fragments over 6 kb by using PCR. To solve these problems, we developed a two-tier GTS workflow by using various DNA assembly methods. In the first-tier, the size of plasmids will be controlled to a range (2 to 10 kb) within which PCR can be reliably used to obtain the needed fragments. We used Q5® High-Fidelity DNA Polymerase from NEB, which has an error rate lower than 4.4×10^{-6} . With this polymerase, the sequencing accuracy of 436 plasmid constructed under GTS reached over 90 % (**Supplementary Figure 6c**).

We further designed the barcode with RE cutting sites to facilitate the assembly in the second-tier, in which Golden Gate assembly (**Figure R4**), conventional RE-based assembly (**Figure R2**) and Yeast assembler (**Figure R5**) can be utilized. By using these methods, we can avoid using PCR in the second-tier. The plasmids obtained in the

second-tier would neither require sequencing confirmation. In the example we used in the previous section - in which a given plasmid can be edited by using REs - the final plasmid would neither require sequencing confirmation. We believe the improved workflow of GTS, thanks to the reviewer, should allow researchers to skip sequencing confirmation in many cases.

We have added two-tier assembly workflows established by using REs in the revised manuscript (line 318-324, line 326-337, and line 405-408).

Figure R4 Golden Gate-based two-tier workflow under GTS. **(a)** Barcodes (5UTR, 3UTR2 and N36) used in the second-tier for assembling operon M1 and M2 fragments (Figure 5a) were modified to contain BsaI recognition sequence, spacer sequence, and 4-nt SE. **(b)** Three plasmids were constructed under GTS in the first-tier assembly. M1 plasmid was assembled by using 4 barcoded fragments (ppsA, aroE, tktA and backbone fragment [Terminator+Amp^R+pAC+Promoter]). M2 plasmid was assembled by using 4 barcoded fragments (aroG, tal, tyrA and backbone fragment [Terminator+Amp^R+pAC+Promoter]). Destination plasmid was assembled by using 3 barcoded fragments (Spec^R, pAC and eGFP expression cassette [Promoter+eGFP+Terminator]). The results of colony PCR and sequencing of the three plasmids are provided on the top of plasmid schematic maps. To generate three 4-nt SEs that can direct final assembly, two module plasmids and one destination plasmid were digested by using BsaI. Then, the enzymatically digested fragments were isolated by using gel electrophoresis, and purified for assembly. The assembly of these three fragments was done at 16 °C for 12 h by using T4 ligase. The primers used for colony PCR are indicated in black arrows. **(c)** Analysis of the enzymatically digested products of three first-tier assembled plasmids. The targeted bands are highlighted in white rectangular boxes, and all three fragments were efficiently liberated from three first-tier plasmids. **(e)** The transformation efficiency of three fragments assembly was evaluated by using CFU/μg DNA. **(f)** Colony PCR using one primer

targeting on M1 region and one targeting on M2 region showed that all the six randomly picked colonies were positive. The correct amplicons are highlighted in white rectangular box. **(g)** Two plasmids extracted from two positive colonies were sequenced to be correct at all three 4-nt SE regions. Other regions of the final plasmid that were covered by sequencing were also correct.

Figure R5 Yeast *in vivo* two-tier workflow under GTS. **(a)** The first-tier assembly was performed to create three plasmids that were used to liberate the fragments for yeast *in vivo* assembly by using REs. PGTP1 was assembled by using 4 barcoded fragments (upstream sequence [UPS], TEF promoter, enhanced yellow fluorescent protein [eYFP] and backbone [AR1+RO1]). PGTP2 was assembled by using 4 barcoded fragments (eYFP, CYC terminator [CYCt], downstream sequence [DWS] and backbone [AR1+RO1]). Yeast backbone (YB) plasmid were constructed by using 3 barcoded fragments (UPS, DWS and Yeast backbone). The results of colony PCR and sequencing are provided on the top of plasmid schematic maps. RO1: *E. coli* replication origin (pMB1); AR1: *E. coli* antibiotic resistance marker (Spec^R); RO2: *E. coli* replication origin (pUC); AR2: *E. coli* antibiotic resistance marker (Amp^R). **(b)** The sequences of barcodes (REC1 and REC2), which include three 8-bp RE recognition sites on each barcode. **(c)** RE recognition sites included in REC1 and REC2. **(d)** The gel electrophoresis analysis of the enzymatically digested products of three first-tier plasmids. The target amplicons are highlighted in white rectangular boxes. The isolated fragments (PGTF1, PGTF2 and YB fragment [YBF]) were purified for further assembly. **(e)** Yeast *in vivo* assembly of 3 fragments. The fragments of UPS, eYFP and DWS were served as homologous arms for guiding the recombination in *S. cerevisiae*. **(f)** The yeast plasmid constructed by assembling three fragments. The primers used in yeast colony PCR are indicated by using black arrows. **(g)** The transformation efficiency of three fragments assembly in yeast was evaluated by using CFU/μg DNA. **(h)** The colonies raised on the plate were exposed under Dark Reader in order to count the number of fluorescent cell and non-fluorescent cell. The fluorescent cell (highlighted in small blue box) and non-fluorescent cell (highlighted in small orange box) were shown in right-side zoom-in figure (highlighted in dark red box). Based on the counting of colonies raised on three plates, the percentage of fluorescent cell and non-fluorescent cell were provided. **(i)** Colony PCR confirmed that all five colonies having fluorescence used as template have correct amplicons, while no correct bands were able to be amplified from the colonies without

fluorescence. (j) One plasmid extracted from a positive yeast colony was sequenced, and the results showed that all the regions covered in sequencing were correct.

Note, the accuracy of plasmid construction is 86% based on Sanger sequencing (lines 168/169); however, here it is not clear whether the errors that occur mostly come from PCR-introduced nucleotide exchanges, or from an erroneous annealing of fragments and barcodes (in wrong order or numbers). Authors should comment on it.

We have excluded almost all unsuccessfully assembled plasmids by using colony PCR, and only colony PCR verified plasmids were extracted for sequencing. The problematic ones that have been observed based on the sequencing were deletion/mutation/insertion/no signal/multiple signals (**Supplementary Figure 5d**), and 72 % of the identified sequence errors were mutation, insertion and deletion outside of assembly junctions, which should have occurred during PCR amplification or be edited by the *E. coli* defense mechanisms.

We have added the description of sequencing errors in the revised manuscript (**line 173-174**).

Is GT assembly suitable for pathway engineering in others hosts? Authors developed GT assembly for pathway engineering in *E. coli*. In principle, however, the method can also be used for building plasmids for other bacterial but also eukaryotic hosts, e.g. *Saccharomyces cerevisiae* (although in yeast, plasmid-based pathway engineering is often suboptimal compared to the engineering of pathways using genome-integrated genes). Authors should shortly discuss the option of extending their method to other organisms. With respect to *S. cerevisiae*, another very prominent host in synthetic biology projects, please note that it is an excellent organism for performing high-throughput *in vivo* cloning at very low cost. The obvious question, therefore, is: would GT assembly be able to compete in yeast with such *in vivo* cloning methods? Authors should discuss this.

It is possible to use GTS for pathway engineering in other microbial hosts. We have developed a two-tier workflow that has also utilized the excellent ability of yeast in assembling DNA fragments (**Figure R5**). GTS would not compete with the yeast-based assembly method, instead it can be used with such method by providing standard parts for it. Based on the fluorescence screening, the assembly accuracy achieved was 83.9 % (**Figure R5**). We are currently working on the assembly of more fragments that can reconstruct a complete biosynthetic pathway (Plasmid- and integration-based) under GTS in *S. cerevisiae*, and believe that this yeast *in vivo*-based workflow could be an efficient tool for researchers to optimize biosynthetic pathway. In another example,

under GTS, we used RE-based workflow (**Figure R2**) to construct a plasmid that should be able to overexpress three genes in *B. subtilis*.

We have added the discussions in the revised manuscript (**line 399-421**).

Other comments:

- Abstract, line 26: authors mention that the method can be used to assemble up to 14 DNA parts in one round of operation. Where is this described in the main text? Actually, authors speak about 2 – 7 fragments in the main text, see also Fig. 1d.

We apologize for causing such a misunderstanding. Fourteen parts referred to seven fragments and seven barcodes. We have revised the sentence in the abstract in the revised manuscript (**line 22-23**).

- Line 51: I think it should read ‘but has NOT yet been adopted widely’.

We have revised sentence in the revised manuscript (**line 40**).

- Line 52: instead of speaking about ‘flaws’ I recommend to speak about ‘limitations’.

We have revised sentence in the revised manuscript (**line 41**).

- Lines 82-84: authors say that ‘most functional DNA sequences can be defined as fragments and/or barcode’. What does this mean with respect to the re-use of fragments (or the offer of standardized fragments by a company, in a kit)? For example, if one lab wants to have a functional DNA part be represented by a ‘fragment’, and another by a ‘barcode’, the reusability idea would be limited to some extent, or? Authors need to comment on this.

Functional DNA sequence can be either fragment or barcode depending on its length. For example, promoter, as a functional biological part, can be either fragment or barcode depending on its length (T7 promoter with LacI repressor expression cassette [~1.7 kb] can be a fragment, while pJ23119 promoter with a length of merely 35 bp can be designed as barcode). Our solution to the hypothetical conflict is to allow both definitions for such short parts. A user would just need to choose the parts that are compatible with his/her own choice of the definition. Such conflict would not occur for a large fraction of parts because they are too large to be defined as barcodes.

- Line 85: the term ‘complementary’ in DNA terms typically means complementarity of DNA sequences (or forward and reverse strands of DNA). Here, however, authors mean

the complementing other half of the barcode. Therefore, I am suggesting to use the term 'complementing'.

We have revised sentence in the revised manuscript (**line 78**).

- Line 132: Fig. 3b is mentioned before Fig. 3a (line 180).

We have modified figure order in the revised manuscript, and revised it accordingly.

- Lines 150/151: authors state that PS bonds were introduced into the fragments AFTER PCR; however, in line 149 they say that the PCR oligonucleotides had PS bonds, meaning that the PCR bonds are introduced into the fragments during the PCR amplification. This needs clarification.

We have corrected the inaccurate description. The PS bonds were introduced DURING the PCR. We have revised sentence in the revised manuscript (**line 141**).

- Line 151: 'simple chemical reaction'; which one? Although it is outlined in the methods part of the manuscript, the reader would benefit from getting a short note here as well.

We have used "Iodine-based cleavage reaction". We have revised sentence in the revised manuscript (**line 146**).

- Lines 163/164: The term 'Aoligo' is introduced here by referring to Fig. 3c, while already Fig. 2a, b mentions Aoligos. If possible, authors should introduce the term earlier in the manuscript.

We have introduced all these terms at the earlier part in the manuscript (**line 108, line 116, line 143 and line 152**).

- Lines 170/171: typically, authors assembled 2 – 7 fragments for plasmid generation, with an accuracy of 86%. Cloning accuracy was similar with different numbers of fragments, and different plasmid sizes over quite a considerable length range (2.4 to 13 kb). However, authors should indicate the lengths of the fragments that were assembled in these experiments, and refer to the respective supplementary table where this is reported in more detail.

We have updated all detailed information in "**Supplementary Table 3**".

- Lines 188 where is the information about the replication origins (RO) and antibiotic resistance markers (AR) given?

We have added the information about RO and AR in the revised manuscript (**line 213**).

- Line 207: the 16 plasmids constructed for engineering E. coli to overproduce tyrosine are not included in Fig. 1d, it seems (as only 8 plasmids are represented there for 6-fragment assemblies). Why not?

We apologize that missing information for the caused misunderstanding. Among the 16 plasmids, half of them used T7 promoter. They were constructed from 6 fragments. The rest of the plasmids used Lac promoter and were constructed from 7 fragments. The difference in the number of fragments was caused by the fact that we did not use LacI repressor expression cassette (encoded in a fragment) when lac promoter was used. We have revised this part in the revised manuscript (**line 231** and **Figure 5a** and **5b** caption).

- Lines 286 – 290: This chapter is not very informative. Authors should better explain what was done; a reference to two supplementary figures is in my opinion not enough.

We have moved this chapter with corresponding descriptions to **Discussion** as an addition feature of GTS, and revised the manuscript accordingly (**line 384-389**).

- Line 631: Is reference 1 correct for the CLIVA method? Note, that in the supplementary data file authors refer to a different reference for CLIVA.

We apologize for the careless mistake. We have corrected it in the revised manuscript.

- I have two questions to Fig. 1d: (FIRST) Authors state they have successfully constructed 370 plasmids using GT assembly (main text, line 166), but in Fig. 1d only about 335 plasmids are represented. Seems, more than 30 plasmids are not shown in the figure. What is the reason for it? (SECOND) The largest plasmids, indicated by the yellow and red color codes, were assembled from 5 fragments, and the medium size plasmids (green) mostly from 4 fragments, while 6- and 7-fragment assemblies were only used for smaller plasmids. Does this mean, that constructing larger plasmids with 6 or 7 fragments does not work? Or did authors simply not test it? Authors should comment on it.

We have corrected this in the revised manuscript (**Figure 1d**) and apologize for the careless mistake caused in processing the data.

We have not tested assembling larger plasmids with 6- or 7-fragment. It has been discussed in the revised manuscript.

- Figure 4: several panels mention plasmids TPP0, TPP18 and PCAP1, however, those are not mentioned in the main text and not explained in the figure legend. Please clarify what these plasmids are/why they were included.

We have clarified this in the manuscript (**line 266-268**).

Reviewers' Comments:

Reviewer #1:

Remarks to the Author:

The authors have addressed all my concerns and the revised manuscript can be accepted for publication.

Reviewer #2:

Remarks to the Author:

As pointed out by the authors, GST does not present a new DNA assembly method; rather, it presents a new DNA assembly standard, that e.g. uses CLIVA for fragment assembly.

The revised manuscript is considerably more accurate than the originally submitted version. In addition, comments of the reviewers were broadly taken into account, and the GST-based assembly standard was extended to align it with other cloning methods.

I am still not convinced about the argument regarding reusability and lower waiting times (to get new oligos). Reusability might only work, as pointed out by the authors, if the entire community (or at least a considerable fraction) adopts the new cloning standard, which is unlikely to happen (in particular considering the ever-dropping prices for oligonucleotides and whole-gene synthesis).

A positive aspect of GTS is that it allows the near scar-less of DNA fragments. However, scar-less assemblies are also possible with other cloning methods, e.g. TAR which employs highly efficient recombination in yeast (*Saccharomyces cerevisiae*). The benefit of GTS over TAR is that it can be entirely performed in *E. coli* without intervening steps in yeast. This may save time and in particular does not require the extra expertise needed to work with yeast (an expertise often not present in labs not working with yeast).

Authors should provide a detailed step-by-step protocol for their GTS assembly as a supplementary file to the manuscript. This would tremendously support other researchers to establish the GTS standard in their labs, as hoped by the authors.

Other points:

- Often, authors do not consecutively refer to figures. E.g., references for Fig. 2 occur in the order Fig. 2a (line 114), Fig. 2e (line 118), Fig. 2c (line 120), Fig. 2f and 2g (line 123), Fig. 2b (line 133), Fig. 2d (line 136). Similarly, Fig. 4a is referenced (line 145) before there is a reference to Fig. 3a (line 184). I think, authors must carefully recheck the manuscript to be in accordance with to the journal's requirements (perhaps also with respect to the referencing to supplementary files, which I did not check).
- Lines 182-185: "We experimentally verified that a panel of type IIS REs can accurately produce the 1-nt SEs after their recognition sequences are included in non-modified Foligos (Fig. 3a and 3b)." I don't think this statement is correct. Authors only cut the DNA, but they did not experimentally confirm that 1-nt sticky ends were produced by the cuts. This must be rephrased to properly reflect the facts.
- Lines 292/293: Authors state that "Two libraries were created. Each contained full combinations of six variants of M1 and M2 (library size: $6 \times 6 = 36$)". As authors have not experimentally proven that indeed the full number of variants has been generated, they should be more cautious with their statement. They better should write: "expected library size: $6 \times 6 = 36$ ".
- Lines 300-302: Authors state "We can easily determine ...". Based on what they actually reported, I would suggest to change the text to "We determined ...".
- Line 311: should read "plasmids using REs".

Response to Reviewers' Comments:

We sincerely appreciate the reviewers' valuable comments. Please find the point-to-point response to reviewers' comments below. [Comments from the reviewers are in black, and our responses are highlighted in blue]

Reviewer #2 (Remarks to the Author):

As pointed out by the authors, GST does not present a new DNA assembly method; rather, it presents a new DNA assembly standard, that e.g. uses CLIVA for fragment assembly.

The revised manuscript is considerably more accurate than the originally submitted version. In addition, comments of the reviewers were broadly taken into account, and the GTS-based assembly standard was extended to align it with other cloning methods.

I am still not convinced about the argument regarding reusability and lower waiting times (to get new oligos). Reusability might only work, as pointed out by the authors, if the entire community (or at least a considerable fraction) adopts the new cloning standard, which is unlikely to happen (in particular considering the ever-dropping prices for oligonucleotides and whole-gene synthesis).

We agree that reusability could be achieved widely if a considerable fraction of community adopts GTS. We have revised the corresponding argument in the revised manuscript to claim the precondition of reusability of oligos under GTS (**line 63-65**).

A positive aspect of GTS is that it allows the near scar-less of DNA fragments. However, scar-less assemblies are also possible with other cloning methods, e.g. TAR which employs highly efficient recombination in yeast (*Saccharomyces cerevisiae*). The benefit of GTS over TAR is that it can be entirely performed in *E. coli* without intervening steps in yeast. This may save time and in particular does not require the extra expertise needed to work with yeast (an expertise often not present in labs not working with yeast).

We hope to highlight again that GTS is a DNA assembly standard. We also believe that it is possible to adopt transformation-associated recombination (TAR)¹ under GTS to construct larger plasmid in yeast. For example, we can barcode a pair of homologous arms, and then assembled them into Bacterial Artificial Chromosome (BAC), Yeast Artificial Chromosome (YAC) or Human Artificial Chromosome (HAC) by using various DNA assembly methods (In-fusion cloning, Gibson, or Golden Gate). The plasmid

constructed under GTS can be further used to isolate large DNA fragment from target genomic DNA through TAR in yeast for further applications (e.g., genome engineering or genome sequencing)¹.

Authors should provide a detailed step-by-step protocol for their GTS assembly as a supplementary file to the manuscript. This would tremendously support other researchers to establish the GTS standard in their labs, as hoped by the authors.

To support other researchers to adopt GTS in their labs, we have provided a detailed step-to-step protocol, and uploaded it to Protocol Exchange as editor suggested (a DOI has been added in the revised manuscript, **line 827**).

Other points:

- Often, authors do not consecutively refer to figures. E.g., references for Fig. 2 occur in the order Fig. 2a (line 114), Fig. 2e (line 118), Fig. 2c (line 120), Fig. 2f and 2g (line 123), Fig. 2b (line 133), Fig. 2d (line 136). Similarly, Fig. 4a is referenced (line 145) before there is a reference to Fig. 3a (line 184). I think, authors must carefully recheck the manuscript to be in accordance with to the journal's requirements (perhaps also with respect to the referencing to supplementary files, which I did not check).

We have revised the inconsecutive reference of figures.

We have carefully checked the manuscript to ensure all the references can be in accordance with the journal's requirements.

- Lines 182-185: "We experimentally verified that a panel of type IIS REs can accurately produce the 1-nt SEs after their recognition sequences are included in non-modified Foligos (Fig. 3a and 3b)." I don't think this statement is correct. Authors only cut the DNA, but they did not experimentally confirm that 1-nt sticky ends were produced by the cuts. This must be rephrased to properly reflect the facts.

We have revised to "We experimentally verified that a panel of type IIS REs could be used to replace the chemical cleavage after their recognition sequences are included in non-modified Foligos (Fig. 4a and 4b). The obtained PCR products were successfully assembled into three plasmids (Supplementary Figure 7), suggesting that the 1-nt SEs on both ends of a fragment were produced efficiently in the one-pot reaction.". (**line 184-191**).

- Lines 292/293: Authors state that "Two libraries were created. Each contained full combinations of six variants of M1 and M2 (library size: $6 \times 6 = 36$)". As authors have not experimentally proven that indeed the full number of variants has been generated, they

should be more cautious with their statement. They better should write: "expected library size: $6 \times 6 = 36$ ".

We have revised it as reviewer suggested (**line 290-291**).

- Lines 300-302: Authors state "We can easily determine ...". Based on what they actually reported, I would suggest to change the text to "We determined ...".

We have revised it as reviewer suggested (**line 297**).

- Line 311: should read "plasmids using REs".

We have revised it as reviewer suggested (**line 308**).

Reference:

1. Kouprina, N., & Larionov, V. (2016). Transformation-associated recombination (TAR) cloning for genomics studies and synthetic biology. *Chromosoma*, 125(4), 621-632.